# Partial Optimal Transport
# with Applications on Positive-Unlabeled Learning

**Laetitia Chapel**
Univ. Bretagne-Sud, CNRS, IRISA
F-56000 Vannes
`laetitia.chapel@irisa.fr`

**Mokhtar Z. Alaya**
LITIS EA4108
University of Rouen Normandy
`mokhtarzahdi.alaya@gmail.com`

**Gilles Gasso**
LITIS EA4108
INSA, University of Rouen Normandy
`gilles.gasso@insa-rouen.fr`

## Abstract

Classical optimal transport problem seeks a transportation map that preserves the total mass between two probability distributions, requiring their masses to be equal. This may be too restrictive in some applications such as color or shape matching, since the distributions may have arbitrary masses and/or only a fraction of the total mass has to be transported. In this paper, we address the partial Wasserstein and Gromov-Wasserstein problems and propose exact algorithms to solve them. We showcase the new formulation in a positive-unlabeled (PU) learning application. To the best of our knowledge, this is the first application of optimal transport in this context and we first highlight that partial Wasserstein-based metrics prove effective in usual PU learning settings. We then demonstrate that partial Gromov-Wasserstein metrics are efficient in scenarii in which the samples from the positive and the unlabeled datasets come from different domains or have different features.

## 1 Introduction

Optimal transport (OT) has been gaining in recent years an increasing attention in the machine learning community, mainly due to its capacity to exploit the geometric property of the samples. Generally speaking, OT is a mathematical tool to compare distributions by computing a transportation mass plan from a source to a target distribution. Distances based on OT are referred to as the Monge-Kantorovich or Wasserstein distances (Villani, 2009) and have been successfully employed in a wide variety of machine learning applications including clustering (Ho et al., 2017), computer vision (Bonneel et al., 2011; Solomon et al., 2015), generative adversarial networks (Arjovsky et al., 2017) or domain adaptation (Courty et al., 2017).

A key limitation of the Wasserstein distance is that it relies on the assumption of aligned distributions, namely they must belong to the same ground space or at least a meaningful distance across domains can be computed. Nevertheless, source and target distributions can be collected under distinct environments, representing different times of collection, contexts or measurements (see Fig. 1, left and right). To get benefit from OT on such heterogeneous distribution settings, one can compute the Gromov-Wasserstein (GW) distance (Sturm, 2006; Mémoli, 2011) to overcome the lack of intrinsic correspondence between the distribution spaces. GW extends Wasserstein by computing a distance between metrics defined within each of the source and target spaces. From a computational point view, it involves a non convex quadratic problem (Peyré and Cuturi, 2019), hard to lift to large scale settings. A remedy to such a heavy computation burden lies in a prevalent approach referred to as

regularized OT (Cuturi, 2013), allowing one to add an entropic regularization penalty to the original problem. Peyré et al. (2016); Solomon et al. (2016) propose the entropic GW discrepancy, that can be solved by Sinkhorn iterations (Cuturi, 2013; Benamou et al., 2015).

A major bottleneck of OT in its traditional formulation is that it requires the two input measures to have the same total probability mass and/or that all the mass has to be transported. This is too restrictive for many applications, such as in color matching or shape registration (Bonneel and Coeurjolly, 2019), since mass changes may occur due to a creation or an annihilation while computing an OT plan. To tackle this limitation, one may employ strategies such as *partial* or *unbalanced transport* (Guittet, 2002; Figalli, 2010; Caffarelli and McCann, 2010). Chizat et al. (2018) propose to relax the marginal constraints of unbalanced total masses using divergences such as Kullback-Leibler or Total Variation, allowing the use of generalized Sinkhorn iterations. Yang and Uhler (2019) generalize this approach to GANs and Lee et al. (2019) present an ADMM algorithm for the relaxed partial OT. Most of all these approaches concentrate on partial-Wasserstein.

This paper deals with exact partial Wasserstein (partial-W) and Gromov-Wasserstein (partial-GW). Some strategies for computing such partial-W require relaxations of the marginals constraints. We rather build our approach upon adding *virtual* or *dummy* points onto the marginals, which is a common practice in OT works. Among the latter, Caffarelli and McCann (2010) attach such points to allow choosing the *maximum distance mass* that can be transported. Pele and Werman (2009) threshold ground distances and send the extra mass to a dummy point to compute a robust EMD distance. Gramfort et al. (2015) consider the case of unnormalized measures and use a dummy point to "fill" the distributions, the extended problem then having both marginals summing to one. More recently, Sarlin et al. (2020) deal with the partial assignment problem by extending the initial problem and fill the ground distance matrix with a single learnable parameter. In this paper, the dummy points are used as a buffer when comparing distributions with different probability masses, allowing partial-W to boil down to solving an extended but standard Wasserstein problem. The main advantage of our approach is that it defines explicitly the mass to be transported and it leads to computing sparse transport plans and hence exact partial-W or -GW distances instead of regularized discrepancies obtained by running Sinkhorn algorithms. Regarding partial-GW, our approach relies on a Frank-Wolfe optimization algorithm (Frank and Wolfe, 1956) that builds on computations of partial-W.

Tackling partial-OT problems that preserve sparsity is motivated by the fact that they are more suitable to some applications such as the Positive-Unlabeled (PU) learning (see Bekker and Davis (2020) for a review) we target in this paper. We shall notice that this is the first application of OT for solving PU learning tasks. In a nutshell, PU classification is a variant of the binary classification problem, in which we have only access to labeled samples from the positive (**Pos**) class in the training stage. The aim is to assign classes to the points of an unlabeled (**Unl**) set which mixes data from both positive and negative classes. Using OT allows identifying the positive points within **Unl**, even when **Pos** and **Unl** samples do not lie in the same space (see Fig. 1).

The paper is organized as follows: we first recall some background on OT. In Section 3, we propose an algorithm to solve an exact partial-W problem, together with a Frank-Wolfe based algorithm to compute the partial-GW solution. After describing in more details the PU learning task and the use of partial-OT to solve it, we illustrate the advantage of partial-GW when the source and the target distributions are collected onto distinct environments. We finally give some perspectives.

**Notations**  $\Sigma_N$ is an histogram of $N$ bins with $\left\{ \boldsymbol{p} \in \mathbb{R}_+^N, \sum_i p_i = 1 \right\}$ and $\delta$ is the Dirac function. Let $\mathbb{1}_n$ be the $n$-dimensional vector of ones. $\langle \cdot, \cdot \rangle_F$ stands for the Frobenius dot product. $|\boldsymbol{p}|$ indicates the length of vector $\boldsymbol{p}$.

## 2  Background on optimal transport

Let $\mathcal{X} = \{\boldsymbol{x}_i\}_{i=1}^n$ and $\mathcal{Y} = \{\boldsymbol{y}_j\}_{j=1}^m$ be two point clouds representing the source and target samples, respectively. We assume two empirical distributions $(\boldsymbol{p}, \boldsymbol{q}) \in \Sigma_n \times \Sigma_m$ over $\mathcal{X}$ and $\mathcal{Y}$,

$$\boldsymbol{p} = \sum_{i=1}^n p_i \delta_{\boldsymbol{x}_i} \quad \text{and} \quad \boldsymbol{q} = \sum_{j=1}^m q_j \delta_{\boldsymbol{y}_j},$$

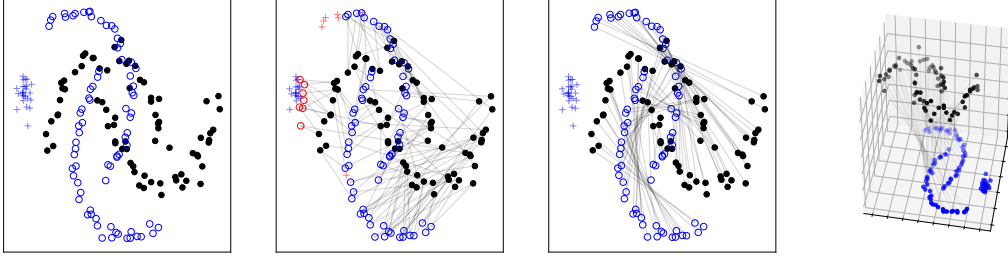

Figure 1: (Left) Source (in black) and target (in blue) samples that have been collected under distinct environments. The source domain contains only positive points ($o$) whereas the target domain contains both positives and negatives ($+$) (Middle left) Partial-W fails to assign correctly all the labels in such context, red symbols indicating wrong assignments (Middle right) Partial-GW recovers the correct labels of the unlabeled samples, with a consistent transportation plan (gray lines), even when the datasets do not live in the same state space (Right).

where $\Sigma_n$ and $\Sigma_m$ are histograms of $|\boldsymbol{p}| = n$ and $|\boldsymbol{q}| = m$ bins respectively. The set of all admissible couplings $\Pi(\boldsymbol{p}, \boldsymbol{q})$ between histograms is given by

$$\Pi(\boldsymbol{p}, \boldsymbol{q}) = \{\boldsymbol{T} \in \mathbb{R}_+^{|\boldsymbol{p}| \times |\boldsymbol{q}|} | \boldsymbol{T}\mathbb{1}_{|\boldsymbol{q}|} = \boldsymbol{p}, \boldsymbol{T}^\top \mathbb{1}_{|\boldsymbol{p}|} = \boldsymbol{q}\},$$

where $\boldsymbol{T} = (T_{ij})_{i,j}$ is a coupling matrix with an entry $T_{ij}$ that describes the amount of mass $p_i$ found at $\boldsymbol{x}_i$ flowing toward the mass $q_j$ of $\boldsymbol{y}_j$.

OT addresses the problem of optimally transporting $\boldsymbol{p}$ toward $\boldsymbol{q}$, given a cost $D_{ij}$ measured as a geometric distance between $\boldsymbol{x}_i$ and $\boldsymbol{y}_j$. More precisely, when the ground cost $\boldsymbol{C} = \mathbf{D}^p = (D_{ij}^p)_{i,j}$ is a distance matrix, the $p$-Wasserstein distance on $\Sigma_n \times \Sigma_m$ at the power of $p$ is defined as:

$$W_p^p(\boldsymbol{p}, \boldsymbol{q}) = \min_{\boldsymbol{T} \in \Pi(\boldsymbol{p}, \boldsymbol{q})} \langle \boldsymbol{C}, \boldsymbol{T} \rangle_F = \min_{\boldsymbol{T} \in \Pi(\boldsymbol{p}, \boldsymbol{q})} \sum_{i=1}^n \sum_{j=1}^m C_{ij} T_{ij}.$$

In some applications, the two distributions are not registered (*i.e.* we can not compute a ground cost between $\boldsymbol{x}_i$ and $\boldsymbol{y}_j$) or do not lie in the same underlying space. The Gromov-Wasserstein distance addresses this bottleneck by extending the Wasserstein distance to such settings, also allowing invariance to translation, rotation or scaling. Informally, it defines the distortion when transporting the whole set of points from one space to another. It relies on intra-domain distance matrices of source $\boldsymbol{C}^s = (C_{ik}^s)_{i,k} = (C^s(\boldsymbol{x}_i, \boldsymbol{x}_k))_{i,k} \in \mathbb{R}_+^{n \times n}$ and target $\boldsymbol{C}^t = (C_{jl}^t)_{j,l} = (C^t(\boldsymbol{y}_j, \boldsymbol{y}_l))_{j,l} \in \mathbb{R}_+^{m \times m}$, and is defined as in Mémoli (2011):

$$GW_p^p(\boldsymbol{p}, \boldsymbol{q}) = \min_{\boldsymbol{T} \in \Pi(\boldsymbol{p}, \boldsymbol{q})} \sum_{i,k=1}^n \sum_{j,l=1}^m \left| C_{ik}^s - C_{jl}^t \right|^p T_{ij} T_{kl}.$$

# 3 Exact Partial Wasserstein and Gromov-Wasserstein distance

We first detail how extending a balanced Wasserstein problem allows solving a partial-Wasserstein one. We then propose a Frank-Wolfe scheme that relies on computing partial-W to solve the partial-GW problem.

## 3.1 Partial Wasserstein distance

The previous OT distances require the two distributions to have the same total probability mass $\|\boldsymbol{p}\|_1 = \|\boldsymbol{q}\|_1$ and that all the mass has to be transported. This may be a problematic assumption where some mass variation or partial mass displacement should be handled. The partial OT problem focuses on transporting only a fraction $0 \le s \le \min(\|\boldsymbol{p}\|_1, \|\boldsymbol{q}\|_1)$ of the mass as cheaply as possible. In that case, the set of admissible couplings becomes

$$\Pi^u(\boldsymbol{p}, \boldsymbol{q}) = \{\boldsymbol{T} \in \mathbb{R}_+^{|\boldsymbol{p}| \times |\boldsymbol{q}|} | \boldsymbol{T}\mathbb{1}_{|\boldsymbol{q}|} \le \boldsymbol{p}, \boldsymbol{T}^\top \mathbb{1}_{|\boldsymbol{p}|} \le \boldsymbol{q}, \mathbb{1}_{|\boldsymbol{p}|}^\top \boldsymbol{T}\mathbb{1}_{|\boldsymbol{q}|} = s\},$$

and the partial-W distance reads as

$$PW_p^p(\boldsymbol{p}, \boldsymbol{q}) = \min_{\boldsymbol{T} \in \Pi^u(\boldsymbol{p}, \boldsymbol{q})} \sum_{i=1}^{n} \sum_{j=1}^{m} \langle \boldsymbol{C}, \boldsymbol{T} \rangle_F.$$

This problem has been studied by (Caffarelli and McCann, 2010; Figalli, 2010); numerical solutions have notably been provided by (Benamou et al., 2015; Chizat et al., 2018) in the entropic-regularized Wasserstein case. We propose here to directly solve the exact partial-W problem by adding *dummy* or *virtual* points $\boldsymbol{x}_{n+1}$ and $\boldsymbol{y}_{m+1}$ (with any features) and extending the cost matrix as follows:

$$\bar{\boldsymbol{C}} = \begin{bmatrix} \boldsymbol{C} & \xi \mathbb{1}_{|\boldsymbol{q}|} \\ \xi \mathbb{1}_{|\boldsymbol{p}|}^{\top} & 2\xi + A \end{bmatrix} \tag{1}$$

in which $A > 0$ and $\xi$ is a fixed positive or nul scalar. When the mass of these dummy points is set such that $p_{n+1} = \|\boldsymbol{q}\|_1 - s$ and $q_{m+1} = \|\boldsymbol{p}\|_1 - s$, computing partial-W distance boils down to solving a unconstrained problem $W_p^p(\bar{\boldsymbol{p}}, \bar{\boldsymbol{q}}) = \min_{\bar{\boldsymbol{T}} \in \Pi(\bar{\boldsymbol{p}}, \bar{\boldsymbol{q}})} \langle \bar{\boldsymbol{C}}, \bar{\boldsymbol{T}} \rangle_F$, where $\bar{\boldsymbol{p}} = [\boldsymbol{p}, \|\boldsymbol{q}\|_1 - s]$ and $\bar{\boldsymbol{q}} = [\boldsymbol{q}, \|\boldsymbol{p}\|_1 - s]$. The intuitive derivation of this equivalent formulation is exposed in Appendix 1.1.

**Proposition 1** *Assume that $A > 0$ and that $\xi$ is a positive or nul scalar, one has*

$$W_p^p(\bar{\boldsymbol{p}}, \bar{\boldsymbol{q}}) - PW_p^p(\boldsymbol{p}, \boldsymbol{q}) = \xi(\|\boldsymbol{p}\|_1 + \|\boldsymbol{q}\|_1 - 2s)$$

*and the optimum transport plan $\boldsymbol{T}^*$ of the partial Wasserstein problem is the optimum transport plan $\bar{\boldsymbol{T}}^*$ of $W_p^p(\bar{\boldsymbol{p}}, \bar{\boldsymbol{q}})$ deprived from its last row and column.*

The proof is postponed to Appendix 1.2.

## 3.2 Partial Gromov-Wasserstein

We are now interested in the partial extension of Gromov-Wasserstein. In the case of a quadratic cost, $p = 2$, the partial-GW problem writes as

$$PGW_2^2(\boldsymbol{p}, \boldsymbol{q}) = \min_{\boldsymbol{T} \in \Pi^u(\boldsymbol{p}, \boldsymbol{q})} \mathcal{J}_{\boldsymbol{C}^s, \boldsymbol{C}^t}(\boldsymbol{T})$$

where

$$\mathcal{J}_{\boldsymbol{C}^s, \boldsymbol{C}^t}(\boldsymbol{T}) = \frac{1}{2} \sum_{i,k=1}^{n} \sum_{j,l=1}^{m} (C_{ik}^s - C_{jl}^t)^2 T_{ij} T_{kl}. \tag{2}$$

The loss function $\mathcal{J}_{\boldsymbol{C}^s, \boldsymbol{C}^t}$ is non-convex and the couplings feasibility domain $\Pi^u(\boldsymbol{p}, \boldsymbol{q})$ is convex and compact. One may expect to introduce virtual points in the GW formulation in order to solve the partial-GW problem. Nevertheless, this strategy is no longer valid as GW involves pairwise distances that do not allow the computations related to the dummy points to be isolated (see Appendix 1.3).

In the following, we build upon a Frank-Wolfe optimization scheme (Frank and Wolfe, 1956) *a.k.a.* the conditional gradient method (Demyanov and Rubinov, 1970). It has received significant renewed interest in machine learning (Jaggi, 2013; Lacoste-Julien and Jaggi, 2015) and in OT community, since it serves as a basis to approximate penalized OT problems (Ferradans et al., 2013; Courty et al., 2017) or GW distances (Peyré et al., 2016; Vayer et al., 2020). Our proposed Frank-Wolfe iterations strongly rely on computing partial-W distances and as such, achieve a sparse transport plan (Ferradans et al., 2013).

Let us first introduce some additional notations. For any tensor $\mathcal{M} = (\mathcal{M}_{ijkl})_{i,j,k,l} \in \mathbb{R}^{n \times n \times m \times m}$, we denote by $\mathcal{M} \circ \boldsymbol{T}$ the matrix in $\mathbb{R}^{n \times m}$ such that its $(i, j)$-th element is defined as

$$(\mathcal{M} \circ \boldsymbol{T})_{i,j} = \sum_{k=1}^{n} \sum_{l=1}^{m} \mathcal{M}_{ijkl} T_{kl}$$

for all $i = 1, \ldots, n, j = 1, \ldots, m$. Introducing the 4-th order tensor $\mathcal{M}(\boldsymbol{C}^s, \boldsymbol{C}^t) = \frac{1}{2}((C_{ik}^s - C_{jl}^t)^2)_{i,j,k,l}$, we notice that $\mathcal{J}_{\boldsymbol{C}^s, \boldsymbol{C}^t}(\boldsymbol{T})$, following Peyré et al. (2016), can be written as

$$\mathcal{J}_{\boldsymbol{C}^s, \boldsymbol{C}^t}(\boldsymbol{T}) = \langle \mathcal{M}(\boldsymbol{C}^s, \boldsymbol{C}^t) \circ \boldsymbol{T}, \boldsymbol{T} \rangle_F.$$

The Frank-Wolfe algorithm for partial-GW is shown in Algorithm 1. Like classical Frank-Wolfe procedure, it is summarized in three steps for each iteration $k$, as detailed below. A theoretical study of the convergence of the Frank-Wolfe algorithm for partial-GW is given in Appendix 2.2, together with a detailed derivation of the line search step (see Appendix 2.1).

**Step1**  Compute a linear minimization oracle over the set $\Pi^u(\boldsymbol{p}, \boldsymbol{q})$, *i.e.*,

$$\widetilde{\boldsymbol{T}}^{(k)} \leftarrow \underset{\boldsymbol{T} \in \Pi^u(\boldsymbol{p}, \boldsymbol{q})}{\operatorname{argmin}} \langle \nabla \mathcal{J}_{\boldsymbol{C}^s, \boldsymbol{C}^t}(\boldsymbol{T}^{(k)}), \boldsymbol{T} \rangle_F, \tag{3}$$

To do so, we solve an extended Wasserstein problem with the ground metric $\nabla \mathcal{J}_{\boldsymbol{C}^s, \boldsymbol{C}^t}(\boldsymbol{T}^{(k)})$ extended as in eq. (1):

$$\bar{\boldsymbol{T}}^{(k)} \leftarrow \underset{\boldsymbol{T} \in \Pi(\bar{\boldsymbol{p}}, \bar{\boldsymbol{q}})}{\operatorname{argmin}} \langle \bar{\nabla} \mathcal{J}_{\boldsymbol{C}^s, \boldsymbol{C}^t}(\boldsymbol{T}^{(k)}), \boldsymbol{T} \rangle_F, \tag{4}$$

and get $\widetilde{\boldsymbol{T}}^{(k)}$ from $\bar{\boldsymbol{T}}^{(k)}$ by removing its last row and column.

**Step2**  Determine optimal step-size $\gamma^{(k)}$ subject to

$$\gamma^{(k)} \leftarrow \underset{\gamma \in [0,1]}{\operatorname{argmin}} \{ \mathcal{J}_{\boldsymbol{C}^s, \boldsymbol{C}^t}((1-\gamma)\boldsymbol{T}^{(k)} + \gamma \widetilde{\boldsymbol{T}}^{(k)}) \}. \tag{5}$$

It can be shown that $\gamma^{(k)}$ can take the following values, with $\boldsymbol{E}^{(k)} = \widetilde{\boldsymbol{T}}^{(k)} - \boldsymbol{T}^{(k)}$:

- if $\langle \mathcal{M}(\boldsymbol{C}^s, \boldsymbol{C}^t) \circ \boldsymbol{E}^{(k)}, \boldsymbol{E}^{(k)} \rangle_F < 0$ we have

$$\gamma^{(k)} = 1$$

- if $\langle \mathcal{M}(\boldsymbol{C}^s, \boldsymbol{C}^t) \circ \boldsymbol{E}^{(k)}, \boldsymbol{E}^{(k)} \rangle_F > 0$ we have

$$\gamma^{(k)} = \min \left( 1, -\frac{\langle \mathcal{M}(\boldsymbol{C}^s, \boldsymbol{C}^t) \circ \boldsymbol{E}^{(k)}, \boldsymbol{T}^{(k)} \rangle_F}{\langle \mathcal{M}(\boldsymbol{C}^s, \boldsymbol{C}^t) \circ \boldsymbol{E}^{(k)}, \boldsymbol{E}^{(k)} \rangle_F} \right)$$

**Step3**  Update $\boldsymbol{T}^{(k+1)} \leftarrow (1 - \gamma^{(k)})\boldsymbol{T}^{(k)} + \gamma^{(k)}\widetilde{\boldsymbol{T}}^{(k)}$.

---

**Algorithm 1** Frank-Wolfe algorithm for partial-GW

---

1: **Input:** Source and target samples: $(\mathcal{X}, \boldsymbol{p})$ and $(\mathcal{Y}, \boldsymbol{q})$, mass $s$, $p = 2$, initial guess $\boldsymbol{T}^{(0)}$
2: Compute cost matrices $\boldsymbol{C}^s$ and $\boldsymbol{C}^t$, build $\bar{\boldsymbol{p}} = [\boldsymbol{p}, \|\boldsymbol{q}\|_1 - s]$ and $\bar{\boldsymbol{q}} = [\boldsymbol{q}, \|\boldsymbol{p}\|_1 - s]$
3: **for** $k = 0, 1, 2, 3, \ldots$ **do**
4:   $\boldsymbol{G}^{(k)} \leftarrow \mathcal{M}(\boldsymbol{C}^s, \boldsymbol{C}^t) \circ \boldsymbol{T}^{(k)}$ // `Compute the gradient` $\nabla \mathcal{J}_{\boldsymbol{C}^s, \boldsymbol{C}^t}(\boldsymbol{T}^{(k)})$
5:   $\bar{\boldsymbol{T}}^{(k)} \leftarrow \operatorname{argmin}_{\boldsymbol{T} \in \Pi(\bar{\boldsymbol{p}}, \bar{\boldsymbol{q}})} \langle \bar{\boldsymbol{G}}^{(k)}, \boldsymbol{T} \rangle_F$ // `Compute partial-W, with` $\bar{\boldsymbol{G}}$ `as in eq. (1)`
6:   Get $\widetilde{\boldsymbol{T}}^{(k)}$ from $\bar{\boldsymbol{T}}^{(k)}$ // `Remove last row and column`
7:   Compute $\gamma^{(k)}$ as in Eq. (5) // `Line-search`
8:   $\boldsymbol{T}^{(k+1)} \leftarrow (1 - \gamma^{(k)})\boldsymbol{T}^{(k)} + \gamma^{(k)}\widetilde{\boldsymbol{T}}^{(k)}$ // `Update`
9: **end for**
10: **Return:** $\boldsymbol{T}^{(k)}$

---

## 4  Optimal transport for PU learning

We hereafter investigate the application of partial optimal transport for learning from Positive and Unlabeled (PU) data. After introducing PU learning, we present how to formulate a PU learning problem into a partial-OT one.

### 4.1  Overview of PU learning

Learning from PU data is a variant of classical binary classification problem, in which the training data consist of only positive points, and the test data is composed of unlabeled positives and negatives. Let $\textbf{Pos} = \{\boldsymbol{x}_i\}_{i=1}^{n_P}$ be the positive samples drawn according to the conditional distribution $p(\boldsymbol{x}|y = 1)$ and $\textbf{Unl} = \{\boldsymbol{x}_i^U\}_{i=1}^{n_U}$ the unlabeled set sampled according to the marginal $p(\boldsymbol{x}) = \pi p(\boldsymbol{x}|y = 1) + (1 - \pi)p(\boldsymbol{x}|y = -1)$. The true proportion of positives, called class prior, is $\pi = p(y = 1)$ and $p(\boldsymbol{x}|y = -1)$ is the distribution of negative samples which are all unlabeled. The goal is to learn a

binary classifier solely using **Pos** and **Unl**. A broad overview of existing PU learning approaches can be seen in (Bekker and Davis, 2020).

Most PU learning methods commonly rely on the selected completely at random (SCAR) assumption (Elkan and Noto, 2008) which assumes that the labeled samples are drawn at random among the positive distribution, independently of their attributes. Nevertheless, this assumption is often violated in real-case scenarii and PU data are often subject to selection biases, *e.g.* when part of the data may be easier to collect. Recently, a less restrictive assumption has been studied: the selected at random (SAR) setting (Bekker and Davis, 2018) which assumes that the positives are labeled according to a subset of features of the samples. Kato et al. (2019) move a step further and consider that the sampling scheme of the positives is such that $p(o = 1|\boldsymbol{x}, y = 1)$ ($o = 1$ means observed label) preserves the ordering induced by the posterior distribution $p(y = 1|\boldsymbol{x})$ over the samples. Other approaches, as in (Hsieh et al., 2019), consider a classical PU learning problem adjuncted with a small proportion of observed negative samples. Those negatives are selected with bias following the distribution $p(\boldsymbol{x}|y = -1)$.

## 4.2 PU learning formulation using partial optimal transport

We propose in this paper to build on partial optimal transport to perform PU learning. In a nutshell, we aim at transporting a mass $s = \pi$ from the unlabeled (source) dataset to the positive (target) one. As such, the transport matrix $\boldsymbol{T}$ should be such that the unlabeled positive points are mapped to the positive samples (as they have similar features or intra-domain distance matrices) while the negatives are discarded (in our context, they are not transported at all).

**Defining the optimal transport point-of-view of PU learning.** More formally, the unlabeled points **Unl** represent the source distribution $\mathcal{X}$ and the positive points **Pos** are the target dataset $\mathcal{Y}$. We set the total probability mass to be transported as the proportion of positives in the unlabeled set, that is $s = \pi$. We look for an optimal transport plan that belongs to the following set of couplings, assuming $n = n_U$, $m = n_P$, $p_i = \frac{1}{n}$ and $q_j = \frac{s}{m}$:

$$\Pi^{PU}(\boldsymbol{p}, \boldsymbol{q}) = \{\boldsymbol{T} \in \mathbb{R}_+^{|\boldsymbol{p}| \times |\boldsymbol{q}|} | \boldsymbol{T}\mathbb{1}_{|\boldsymbol{q}|} = \{\boldsymbol{p}, 0\}, \boldsymbol{T}^\top \mathbb{1}_{|\boldsymbol{p}|} \leq \boldsymbol{q}, \mathbb{1}_{|\boldsymbol{p}|}^\top \boldsymbol{T}\mathbb{1}_{|\boldsymbol{q}|} = s\}, \quad (6)$$

in which $\boldsymbol{T}\mathbb{1}_{|\boldsymbol{q}|} = \{\boldsymbol{p}, 0\}$ means that $\sum_{j=1}^m T_{ij} = p_i$ exactly or $0$, $\forall i$ to avoid matching part of the mass of an unlabeled negative with a positive. This set is not empty as long as $s \mod p_i = 0$. The problem that we aim at solving is the following:

$$PUW_p^p(\boldsymbol{p}, \boldsymbol{q}) = \min_{\boldsymbol{T} \in \Pi^{PU}(\boldsymbol{p}, \boldsymbol{q})} \sum_{i=1}^n \sum_{j=1}^m C_{ij} T_{ij}.$$

Though the positive samples **Pos** are assumed easy to label, their features may be corrupted with noise or they may be mislabeled. Let assume $0 \leq \alpha \leq 1 - s$, the noise level.

**Solving the PU problem.** To enforce the condition $\boldsymbol{T}\mathbb{1}_{|\boldsymbol{q}|} = \{\boldsymbol{p}, 0\}$, we adopt a regularized point of view of the partial-OT problem as in Courty et al. (2017) and we solve the following problem:

$$\bar{\boldsymbol{T}}^* = \operatorname*{argmin}_{\bar{\boldsymbol{T}} \in \Pi(\bar{\boldsymbol{p}}, \bar{\boldsymbol{q}})} \sum_{i=1}^{n+1} \sum_{j=1}^{m+1} \bar{C}_{ij} \bar{T}_{ij} + \eta \Omega(\bar{\boldsymbol{T}}) \quad (7)$$

where $p_i = \frac{1-\alpha}{n}$, $q_j = \frac{s+\alpha}{m}$, $\bar{\boldsymbol{p}}, \bar{\boldsymbol{q}}, \bar{C}_{ij}$ are defined as in Section 3.1, $\eta \geq 0$ is a regularization parameter and $\alpha$ is the percentage of **Pos** that we assume to be noisy (that is to say we do not want to map them to a point of **Unl**). We choose $\Omega(\bar{\boldsymbol{T}}) = \sum_{i=1}^n \left( \|\bar{\boldsymbol{T}}_{i(:m)}\|_2 + \|\bar{\boldsymbol{T}}_{i(m+1)}\|_2 \right)$ where $\bar{\boldsymbol{T}}_{i(:m)}$ is a vector that contains the entries of the $i^{\text{th}}$ row of $\bar{\boldsymbol{T}}$ associated to the first $m$ columns. This group-lasso regularization leads to a sparse transportation map and enforces each of the **Unl** samples $\boldsymbol{x}_i$ to be mapped to only the **Pos** samples or to the dummy point $\boldsymbol{y}_{m+1}$. An illustration is provided in Appendix 5. When partial-GW is involved, we use this regularized-OT in the step $(i)$ of the Frank-Wolfe algorithm.

We can establish that solving problem (7) provides the solution to PU learning using partial-OT.

**Proposition 2** *Assume that $A > 0$, $\xi$ is a constant, there exists a large $\eta > 0$ such that:*

$$W_p^{*p}(\bar{\boldsymbol{p}}, \bar{\boldsymbol{q}}) - PUW_p^p(\boldsymbol{p}, \boldsymbol{q}) = \xi(1 - s).$$

*where $W_p^{*p}(\bar{\boldsymbol{p}}, \bar{\boldsymbol{q}}) = \sum_{i=1}^{n+1} \sum_{j=1}^{m+1} \bar{C}_{ij} \bar{T}_{ij}^*$ with $\bar{\boldsymbol{T}}$ solution of eq. (7).*

The proof is postponed to Appendix 3.

## 5 Experiments

### 5.1 Experimental design

We illustrate the behavior of partial-W and -GW on real datasets in a PU learning context. First, we consider a SCAR assumption, then a SAR one and finally a more general setting, in which the underlying distributions of the samples come from different domains, or do not belong to the same metric space. Algorithm 1 has been implemented and is avalaible on the Python Optimal Transport (POT) toolbox (Flamary and Courty, 2017).

Following previous works (Kato et al., 2019; Hsieh et al., 2019), we assume that the class prior $\pi$ is known throughout the experiments; otherwise, it can be estimated from $\{\boldsymbol{x}_i\}_{i=1}^{n_P}$ and $\{\boldsymbol{x}_i^U\}_{i=1}^{n_U}$ using off-the-shelf methods, *e.g.* Zeiberg and Radivojac (2020); Plessis et al. (2017); Jain and Radivojac (2016). For both partial-W and partial-GW, we choose $p = 2$ and the cost matrices $\boldsymbol{C}$ are computed using Euclidean distance.

We carry experiments on real-world datasets under the aforementioned scenarii. We rely on six datasets `Mushrooms`, `Shuttle`, `Pageblocks`, `USPS`, `Connect-4`, `Spambase` from the UCI repository[1] (following Kato et al. (2019)'s setting) and `colored MNIST` (Arjovsky et al., 2019) to illustrate our method in SCAR and SAR settings respectively. We also consider the `Caltech office` dataset, which is a common application of domain adaptation (Courty et al., 2017) to explore the effectiveness of our method on heterogeneous distribution settings.

Whenever they contain several classes, these datasets are converted into binary classification problems following Kato et al. (2019), and the positives are the samples that belong to the $y = 1$ class. For UCI and `colored MNIST` datasets, we randomly draw $n_P = 400$ positive and $n_U = 800$ unlabeled points among the remaining data. As the `Caltech office` datasets are smaller, we choose $n_P = 100$ and $n_U = 100$ in that context. To ease the presentation, we report here the results with class prior $\pi$ set as the true proportion of positive class in the dataset and provide in Appendix 6.3 additional results when varying $s$. We ran the experiments 10 times and report the mean accuracy rate (standard deviations are shown in Appendix 6.1). We test 2 levels of noise in **Pos**: $\alpha = 0$ or $\alpha = 0.025$, fix $\xi = 0$, $A = \max(\boldsymbol{C})$ and choose a large $\eta = 10^6$.

For the experiments, we consider unbiased PU learning method (denoted by PU in the sequel) (Du Plessis et al., 2014) and the most recent and effective method to address PU learning with a selection bias (called PUSB below) that tries to weaken the SCAR assumption (Kato et al., 2019). Whenever possible (that is to say when source and target samples share the same features), we compare our approaches P-W and P-GW with PU and PUSB; if not, we are not aware of any competitive PU learning method able to handle different features in **Pos** and **Unl**. The GW formulation is a non convex problem and the quality of the solution is highly dependent on the initialization. We explore several initializations of the transport matrix for P-GW and report the results that yield to the lowest partial OT-distance (see Appendix 4 for details).

### 5.2 Partial-W and partial-GW in a PU learning under a SCAR assumption

Under SCAR, the **Pos** dataset and the positives in **Unl** are assumed independently and identically drawn according to the distribution $p(\boldsymbol{x}|y = 1)$ from a set of positive points. We experiment on the UCI datasets and Table 1 (top) summarizes our findings. Except for `Connect-4` and `spambase`, partial-W has similar results or consistently outperforms PU and PUSB. Including some noise has little impact on the results, except for the `connect-4` dataset. Partial-GW has competitive results, showing that relying on intra-domain matrices may allow discriminating the classes. It nevertheless

under-performs relatively to partial-W, as the distance matrix $C$ between **Pos** and **Unl** is more informative than only relying on intra-domain matrices.

Table 1: Average accuracy rates on various datasets. (G)-PW 0 indicates no noise and (G)P-W 0.025 stands for a noise level of $\alpha = 0.025$. Best values are indicated boldface.

| DATASET | $\pi$ | PU | PUSB | P-W 0 | P-W 0.025 | P-GW 0 | P-GW 0.025 |
|---|---|---|---|---|---|---|---|
| MUSHROOMS | 0.518 | 91.1 | 90.8 | 96.3 | **96.4** | 95.0 | 93.1 |
| SHUTTLE | 0.786 | 90.8 | 90.3 | **95.8** | 94.0 | 94.2 | 91.8 |
| PAGEBLOCKS | 0.898 | 92.1 | 90.9 | **92.2** | 91.6 | 90.9 | 90.8 |
| USPS | 0.167 | 95.4 | 95.1 | **98.3** | 98.1 | 94.9 | 93.3 |
| CONNECT-4 | 0.658 | **65.6** | 58.3 | 55.6 | 61.7 | 59.5 | 60.8 |
| SPAMBASE | 0.394 | **84.3** | 84.1 | 78.0 | 76.4 | 70.2 | 71.2 |
| ORIGINAL MNIST | 0.1 | 97.9 | 97.8 | **98.8** | 98.6 | 98.2 | 97.9 |
| COLORED MNIST | 0.1 | 87.0 | 80.0 | 91.5 | 91.5 | 97.3 | **98.0** |
| SURF C→SURF C | 0.1 | 89.3 | 89.4 | 90.0 | **90.2** | 87.2 | 87.0 |
| SURF C→SURF A | 0.1 | **87.7** | 85.6 | 81.6 | 81.8 | 85.6 | 85.6 |
| SURF C→SURF W | 0.1 | 84.4 | 80.5 | 82.2 | 82.0 | **85.6** | 85.0 |
| SURF C→SURF D | 0.1 | 82.0 | 83.2 | 80.0 | 80.0 | 87.6 | **87.8** |
| DECAF C→DECAF C | 0.1 | 93.9 | **94.8** | 94.0 | 93.2 | 86.4 | 87.2 |
| DECAF C→DECAF A | 0.1 | 80.5 | 82.2 | 80.2 | 80.2 | **89.2** | 88.8 |
| DECAF C→DECAF W | 0.1 | 82.4 | 83.8 | 80.2 | 80.2 | **89.2** | 88.6 |
| DECAF C→DECAF D | 0.1 | 82.6 | 83.6 | 80.8 | 80.6 | **94.2** | 93.2 |

## 5.3 Experiments under a SAR assumption

The SAR assumption supposes that **Pos** is drawn according to some features of the samples. To implement such a setting, we inspire from (Arjovsky et al., 2019) and we construct a colored version of MNIST: each digit is colored, either in green or red, with a probability of $90\%$ to be colored in red. The probability to label a digit $y = 1$ as positive depends on its color, with only green $y = 1$ composing the positive set. The **Unl** dataset is then mostly composed of red digits. Results under this setting are provided in Table 1 (middle). When we consider a SCAR scenario, partial-W exhibits the best performance. However, its effectiveness highly drops when a covariate shift appears in the distributions $p(\boldsymbol{x}|y = 1)$ of the **Pos** and **Unl** datasets as in this SAR scenario. On the opposite, partial-GW allows maintaining a comparable level of accuracy as the discriminative information are preserved in intra-domain distance matrices.

## 5.4 Partial-W and -GW in a PU learning with different domains and/or feature spaces

To further validate the proposed method in a different context, we apply partial-W and partial-GW to a domain adaptation task. We consider the Caltech Office dataset, that consists of four domains: Caltech 256 (C) (Griffin et al., 2007), Amazon (A), Webcam (W) and DSLR (D) (Saenko et al., 2010). There exists a high inter-domain variability as the objects may face different illumination, orientation *etc*. Following a standard protocol, each image of each domain is described by a set of SURF features (Saenko et al., 2010) consisting of a normalized 800-bins histogram, and by a set of DECAF features (Donahue et al., 2014), that are 4096-dimensional features extracted from a neural network. The **Pos** dataset consists of images from Caltech 256. The unlabeled samples are formed by the Amazon, Webcam, DSLR images together with the Caltech 256 images that are not included in **Pos**. We perform a PCA to project the data onto $d = 10$ dimensions for the SURF features and $d = 40$ for the DECAF ones.

We first investigate the case where the objects are represented by the same features but belong to the same or different domains. Results are given in Table 1 (bottom). For both features, we first notice that PU and PUSB have similar performances than partial-W when the domains are the same. As soon as the two domains differ, partial-GW exhibits the best performances, suggesting that it is able to capture some domain shift. We then consider a scenario where the source and target objects are described by different features (Table 2). In that case, only partial-GW is applicable and its performances suggest that it is able to efficiently leverage on the discriminative information conveyed in the intra-domain similarity matrices, especially when using SURF features to make predictions based on DECAF ones.

Table 2: Average accuracy rates on domain adaptation scenarii described by different features. As there is little difference between the obtained results when considering the two levels of noise, we report performances only for P-GW 0.

| SCENARIO | *=C | *=A | *=W | * = D |
|---|---|---|---|---|
| SURF C→DECAF * | 87.0 | 94.4 | 94.4 | 97.4 |
| DECAF C→SURF * | 85.0 | 83.2 | 83.8 | 82.8 |

# 6 Conclusion and future work

In this paper, we build on partial-W and -GW distances to solve a PU learning problem. We propose a scheme relying on iterations of a Frank-Wolfe algorithm to compute a partial-GW solution, in which each iteration requires solving a partial-W problem that is derived from the solution of an extended Wassertein problem. We show that those distances compete and sometimes outperform the state-of-the-art PU learning methods, and that partial-GW allows remarkable improvements when the underlying spaces of the positive and unlabeled datasets are distinct or even unregistered.

While considering only features (with partial-W) or intra-domain distances (with partial-GW), this work can be extended to define a partial-Fused Gromov-Wasserstein distance (Vayer et al., 2020) that can combines both aspects. Another line of work will also focus on lowering the computational complexity by using *sliced* partial-GW, building on existing works on *sliced* partial-W (Bonneel and Coeurjolly, 2019) and *sliced* GW (Vayer et al., 2019). Regarding the application view point, we envision a potential use of the approach to subgraph matching (Kriege and Mutzel, 2012) or PU learning on graphs (Zhao et al., 2011) as GW has been proved to be effective to compare structured data such as graphs. In addition, we also target applications such as detecting out-of-distributions examples or open-set domain adaptation (Saito et al., 2018). Finally, we plan to derive an extension of this work to PU learning in which the proportion of positives in the dataset will be estimated in a unified optimal transport formulation, building on results of GW-based test of isomorphism between distributions (Brécheteau, 2019).

# Broader impact

This work does not present any significant societal, environnemental or ethical consequence.

# Acknowledgments

This work is partially funded through the projects OATMIL ANR-17-CE23-0012, MULTISCALE ANR-18-CE23-0022-01 and RAIMO ANR-20-CHIA-0021-01.

## Footnotes

[1]https://archive.ics.uci.edu/ml/datasets.php

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
