[Supplementary Material]

# Supplementary Material for "Partial Optimal Transport with Applications on Positive-Unlabeled Learning'

**Laetitia Chapel**
Univ. Bretagne-Sud, CNRS, IRISA
F-56000 Vannes
laetitia.chapel@irisa.fr

**Mokhtar Z. Alaya**
LITIS EA4108
University of Rouen Normandy
mokhtarzahdi.alaya@gmail.com

**Gilles Gasso**
LITIS EA4108
INSA, University of Rouen Normandy
gilles.gasso@insa-rouen.fr

## 1 Partial Optimal Transport with dummy points

The proof involves 3 steps:

1. we first justify the definition of $\bar{\boldsymbol{p}}$ and $\bar{\boldsymbol{q}}$ in the extended problem formulation, and show that $\bar{T}_{(n+1)(m+1)}$ should be equal to zero in order to have an equivalence between the original and the extended constraint set;

2. we then show that, for an optimal $\bar{\boldsymbol{T}}^*$, we have $\bar{T}^*_{(n+1)(m+1)} = 0$ if $\bar{C}_{(n+1)(m+1)} > 2\xi$;

3. we finally show that the solution of the extended Wasserstein problem $\bar{\boldsymbol{T}}^*$ deprived from its last row and column and the one of the partial-Wasserstein one $\boldsymbol{T}^*$ are the same, and we show that $W^p_p(\bar{\boldsymbol{p}}, \bar{\boldsymbol{q}}) - PW^p_p(\boldsymbol{p}, \boldsymbol{q}) = \xi(\|\boldsymbol{p}\|_1 + \|\boldsymbol{q}\|_1 - 2s)$.

### 1.1 Equivalence between the constraint set of the extended Wasserstein problem and the partial-W problem

Let recall the formulation of partial OT problem that aims to transport only a fraction $0 \le s \le \min(\|\boldsymbol{p}\|_1, \|\boldsymbol{q}\|_1)$ of the mass as cheaply as possible. In that case, the problem to solve is:

$$\min_{\boldsymbol{T} \in \Pi^u(\boldsymbol{p}, \boldsymbol{q})} \langle \boldsymbol{C}, \boldsymbol{T} \rangle_F$$

with the constraint set:

$$\Pi^u(\boldsymbol{p}, \boldsymbol{q}) = \left\{ \boldsymbol{T} \in \mathbb{R}^{|\boldsymbol{p}| \times |\boldsymbol{q}|}_+ | \boldsymbol{T} \mathbb{1}_{|\boldsymbol{q}|} \le \boldsymbol{p}, \boldsymbol{T}^\top \mathbb{1}_{|\boldsymbol{p}|} \le \boldsymbol{q}, \mathbb{1}^\top_{|\boldsymbol{p}|} \boldsymbol{T} \mathbb{1}_{|\boldsymbol{q}|} = s \right\}. \tag{1}$$

To express this as a standard discrete Kantorovitch optimal transport problem, we get rid of the inequality constraints by re-writting (using standard tricks of linear programming):

$$\boldsymbol{T} \mathbb{1}_{|\boldsymbol{q}|} + \boldsymbol{b} = \boldsymbol{p}$$
$$\boldsymbol{T}^\top \mathbb{1}_{|\boldsymbol{p}|} + \boldsymbol{a} = \boldsymbol{q}$$

with $b \in \mathbb{R}_+^{|p|}$ and $a \in \mathbb{R}_+^{|q|}$ two unknown vectors. Using the fact that $\mathbb{1}_{|p|}^\top T \mathbb{1}_{|q|} = s$, we get then the following equality constraint for marginal $q$:

$$\mathbb{1}_{|p|}^\top T \mathbb{1}_{|q|} + \mathbb{1}_{|p|}^\top b = \mathbb{1}_{|p|}^\top p$$
$$s + \mathbb{1}_{|p|}^\top b = \|p\|_1$$

and for marginal $p$

$$\mathbb{1}_{|p|}^\top T \mathbb{1}_{|q|} + a^\top \mathbb{1}_{|q|} = q^\top \mathbb{1}_{|q|}$$
$$s + a^\top \mathbb{1}_{|q|} = \|q\|_1$$

The relations $s + a^\top \mathbb{1}_{|q|} = \|p\|_1$ and $s + \mathbb{1}_{|p|}^\top b = \|q\|_1$ take into account the constraint related to the transported mass $s$ and we establish in subsection 1.2.1 of the supplementary material that this mass is preserved whenever we solve the extended problem without the explicit constraint $\mathbb{1}_{|p|}^\top T \mathbb{1}_{|q|} = s$. Gathering these elements leads to this equivalent formulation

$$\min_{T \in \Pi^u(p,q)} \langle C, T \rangle_F$$

with $\Pi^u(p, q)$ re-expressed as

$$\Pi^u(p, q) = \{ T \in \mathbb{R}_+^{|p| \times |q|} \mid T \mathbb{1}_{|q|} + b = p, T^\top \mathbb{1}_{|p|} + a = q, \mathbb{1}_{|p|}^\top T \mathbb{1}_{|q|} = s,$$
$$a \in \mathbb{R}_+^{|q|}, a^\top \mathbb{1}_{|q|} = \|q\|_1 - s, b \in \mathbb{R}_+^{|p|}, \mathbb{1}_{|p|}^\top b = \|p\|_1 - s \}.$$

By defining the following augmented matrices and vectors

$$\bar{T} = \begin{bmatrix} T & b \\ a^\top & \beta \end{bmatrix}, \quad \bar{p} = \begin{bmatrix} p \\ \|q\|_1 - s \end{bmatrix}, \quad \text{and} \quad \bar{q} = \begin{bmatrix} q \\ \|p\|_1 - s \end{bmatrix}$$

we get this compact formulation

$$\Pi^e(\bar{p}, \bar{q}) = \{ \bar{T} \in \mathbb{R}_+^{|\bar{p}| \times |\bar{q}|} \mid \bar{T} \mathbb{1}_{|\bar{q}|} = \bar{p}, \bar{T}^\top \mathbb{1}_{|\bar{p}|} = \bar{q}, \beta = 0 \}. \tag{2}$$

In the following, we show how solving a Wasserstein problem under the constraint set (2) and how recovering the solution of the original partial problem (with the constraint set 1) from it.

## 1.2  Proof of Proposition 1

Let us denote $\bar{T}^*$ the optimal coupling of the extended problem

$$\bar{T}^* = \operatorname*{argmin}_{\bar{T} \in \Pi(\bar{p},\bar{q})} \sum_{i=1}^{n+1} \sum_{j=1}^{m+1} \bar{C}_{ij} \bar{T}_{ij}$$

and recall that we set

$$\bar{C} = \begin{bmatrix} C & \xi \mathbf{1}_m \\ \xi \mathbf{1}_n^\top & 2\xi + A \end{bmatrix}$$

with $A > 0$.

### 1.2.1  We first check that $\mathbb{1}_n^\top \bar{T}^{\backslash *} \mathbb{1}_m = s$ when $\bar{T}^*_{(n+1)(m+1)} = 0$

Assume $\bar{T}^{\backslash *}$ is the matrix $\bar{T}^*$ with the last row and column removed. Let us first suppose that $\bar{T}^*_{(n+1)(m+1)} = 0$. As a consequence and because of the constraints on the marginals, we have

$$\sum_{j=1}^{m} \bar{T}^*_{(n+1)j} = \|q\|_1 - s$$

and

$$\sum_{i=1}^{n} \bar{T}^*_{i(m+1)} = \|p\|_1 - s.$$

We can also easily see that

$$\mathbb{1}_{n+1}^\top \bar{\boldsymbol{T}}^* \mathbb{1}_{m+1} = \|\boldsymbol{p}\|_1 + \|\boldsymbol{q}\|_1 - s$$

as $\|\bar{\boldsymbol{p}}\|_1 = \|\bar{\boldsymbol{q}}\|_1 = \|\boldsymbol{p}\|_1 + \|\boldsymbol{q}\|_1 - s$. This implies that

$$
\begin{aligned}
\mathbb{1}_n^\top \bar{\boldsymbol{T}}^{\backslash *} \mathbb{1}_m &= \mathbb{1}_{n+1}^\top \bar{\boldsymbol{T}}^* \mathbb{1}_{m+1} - \sum_{j=1}^m \bar{T}^*_{(n+1)j} - \sum_{i=1}^n \bar{T}^*_{i(m+1)} - \bar{T}^*_{(n+1)(m+1)} \\
&= (\|\boldsymbol{p}\|_1 + \|\boldsymbol{q}\|_1 - s) - (\|\boldsymbol{q}\|_1 - s) - (\|\boldsymbol{p}\|_1 - s) - 0 \\
&= s.
\end{aligned}
$$

Hence, we have established that $\mathbb{1}_n^\top \bar{\boldsymbol{T}}^{\backslash *} \mathbb{1}_m = s$.

### 1.2.2  Let show that $\mathbb{1}_n^\top \bar{\boldsymbol{T}}^{\backslash *} \mathbb{1}_m = s + \bar{T}^*_{(n+1)(m+1)}$ when $\bar{T}^*_{(n+1)(m+1)} \neq 0$

Let us now suppose that $T^*_{(n+1)(m+1)} \neq 0$. This implies that

$$
\begin{aligned}
\mathbb{1}_n^\top \bar{\boldsymbol{T}}^{\backslash *} \mathbb{1}_m &= \mathbb{1}_{n+1}^\top \bar{\boldsymbol{T}}^* \mathbb{1}_{m+1} - \sum_{j=1}^{m+1} \bar{T}^*_{(n+1)j} - \sum_{i=1}^{n+1} \bar{T}^*_{i(m+1)} + \bar{T}^*_{(n+1)(m+1)} \\
&= (\|\boldsymbol{p}\|_1 + \|\boldsymbol{q}\|_1 - s) - (\|\boldsymbol{q}\|_1 - s) - (\|\boldsymbol{p}\|_1 - s) + \bar{T}^*_{(n+1)(m+1)} \\
&= s + \bar{T}^*_{(n+1)(m+1)}
\end{aligned}
$$

### 1.2.3  We now show that $\bar{T}^*_{(n+1)(m+1)} = 0$ when $\bar{C}_{(n+1)(m+1)} > 2\xi$

We have

$$
\begin{aligned}
\bar{\boldsymbol{T}}^* &= \operatorname*{argmin}_{\bar{\boldsymbol{T}} \in \Pi(\bar{\boldsymbol{p}}, \bar{\boldsymbol{q}})} \sum_{i=1}^{n+1} \sum_{j=1}^{m+1} \bar{C}_{ij} \bar{T}_{ij} \\
&= \operatorname*{argmin}_{\bar{\boldsymbol{T}} \in \Pi(\bar{\boldsymbol{p}}, \bar{\boldsymbol{q}})} \sum_{i=1}^n \sum_{j=1}^m \bar{T}_{ij} \bar{C}_{ij} + \sum_{i=1}^n \bar{T}_{i(m+1)} \bar{C}_{i(m+1)} + \sum_{j=1}^m \bar{T}_{(n+1)j} \bar{C}_{(n+1)j} \\
&\qquad + \bar{T}_{(n+1)(m+1)} \bar{C}_{(n+1)(m+1)} \\
&= \operatorname*{argmin}_{\bar{\boldsymbol{T}} \in \Pi(\bar{\boldsymbol{p}}, \bar{\boldsymbol{q}})} \sum_{i=1}^n \sum_{j=1}^m \bar{T}_{ij} C_{ij} + \xi \sum_{i=1}^n \bar{T}_{i(m+1)} + \xi \sum_{j=1}^m \bar{T}_{(n+1)j} + \bar{T}_{(n+1)(m+1)} \bar{C}_{(n+1)(m+1)} \\
&= \operatorname*{argmin}_{\bar{\boldsymbol{T}} \in \Pi(\bar{\boldsymbol{p}}, \bar{\boldsymbol{q}})} \sum_{i=1}^n \sum_{j=1}^m \bar{T}_{ij} C_{ij} + \xi(\|\boldsymbol{p}\|_1 - s - \bar{T}_{(n+1)(m+1)}) + \xi(\|\boldsymbol{q}\|_1 - s - \bar{T}_{(n+1)(m+1)}) \\
&\qquad + \bar{T}_{(n+1)(m+1)} \bar{C}_{(n+1)(m+1)} \\
&= \operatorname*{argmin}_{\bar{\boldsymbol{T}} \in \Pi(\bar{\boldsymbol{p}}, \bar{\boldsymbol{q}})} \sum_{i=1}^n \sum_{j=1}^m \bar{T}_{ij} C_{ij} + \xi(\|\boldsymbol{p}\|_1 + \|\boldsymbol{q}\|_1 - 2s) + (\bar{C}_{(n+1)(m+1)} - 2\xi) \bar{T}_{(n+1)(m+1)}.
\end{aligned}
$$

Let us now suppose that $\bar{T}^*_{(n+1)(m+1)} > 0$. Let us suppose that there exists a coupling $\boldsymbol{\Gamma}$ that belongs to the admissible constraint set $\Pi(\bar{\boldsymbol{p}}, \bar{\boldsymbol{q}})$ such that $\Gamma_{(n+1)(m+1)} = 0$. We then have:

$$
\sum_{i=1}^n \sum_{j=1}^m \bar{T}^*_{ij} C_{ij} + \xi(\|\boldsymbol{p}\|_1 + \|\boldsymbol{q}\|_1 - 2s) + (\bar{C}_{(n+1)(m+1)} - 2\xi) \bar{T}^*_{(n+1)(m+1)}
$$

$$
\leq \sum_{i=1}^n \sum_{j=1}^m \Gamma_{ij} C_{ij} + \xi(\|\boldsymbol{p}\|_1 + \|\boldsymbol{q}\|_1 - 2s)
$$

that we can rewrite as

$$\sum_{i=1}^{n}\sum_{j=1}^{m}\bar{T}_{ij}^{*}C_{ij} + (\bar{C}_{(n+1)(m+1)} - 2\xi)\bar{T}_{(n+1)(m+1)}^{*} \leq \sum_{i=1}^{n}\sum_{j=1}^{m}\Gamma_{ij}C_{ij}. \text{ Then,}$$

$$(\bar{C}_{(n+1)(m+1)} - 2\xi)\bar{T}_{(n+1)(m+1)}^{*} \leq \sum_{i=1}^{n}\sum_{j=1}^{m}\Gamma_{ij}C_{ij} - \sum_{i=1}^{n}\sum_{j=1}^{m}T_{ij}^{*}C_{ij}$$

We know that $\sum_{i=1}^{n}\sum_{j=1}^{m}\bar{T}_{ij}^{*} = s + \bar{T}_{(n+1)(m+1)}^{*}$ and that $\sum_{i=1}^{n}\sum_{j=1}^{m}\Gamma_{ij} = s$ by construction. We can then write

$$\sum_{i=1}^{n}\sum_{j=1}^{m}\Gamma_{ij} = \sum_{i=1}^{n}\sum_{j=1}^{m}\bar{T}_{ij}^{*} - \bar{T}_{(n+1)(m+1)}^{*}$$

and it implies that we can build a coupling $\mathbf{\Gamma}$ such that

$$\sum_{i=1}^{n}\sum_{j=1}^{m}\Gamma_{ij}C_{ij} \leq \sum_{i=1}^{n}\sum_{j=1}^{m}\bar{T}_{ij}^{*}C_{ij}$$

Such a coupling can be obtained by moving some of the mass $\bar{T}_{ij}^{*} > 0$ ($i$ and $j$s can be chosen randomly), with $i \leq n$ and $j \leq m$, towards the last row and column, such that we move a mass $\bar{T}_{(n+1)(m+1)}^{*}$, ending up with $\sum_{i=1}^{n}\sum_{j=1}^{m}\Gamma_{ij} = s$. It is straighforward to see that, by doing so, we ensure that $\mathbf{\Gamma}$ remains an admissible coupling (see fig. 1 for an illustration). Then,

$$(\bar{C}_{(n+1)(m+1)} - 2\xi)\bar{T}_{(n+1)(m+1)}^{*} \leq \sum_{i=1}^{n}\sum_{j=1}^{m}\Gamma_{ij}C_{ij} - \sum_{i=1}^{n}\sum_{j=1}^{m}\bar{T}_{ij}^{*}C_{ij}$$

$$(\bar{C}_{(n+1)(m+1)} - 2\xi)\bar{T}_{(n+1)(m+1)}^{*} \leq \sum_{i=1}^{n}\sum_{j=1}^{m}\Gamma_{ij}C_{ij} - \sum_{i=1}^{n}\sum_{j=1}^{m}\Gamma_{ij}C_{ij}$$

$$(\bar{C}_{(n+1)(m+1)} - 2\xi)\bar{T}_{(n+1)(m+1)}^{*} \leq 0$$

Setting $\bar{C}_{(n+1)(m+1)} > 2\xi$ contradicts the initial hypothesis, and then we can conclude that $\bar{T}_{(n+1)(m+1)}^{*} = 0$ when $\bar{C}_{(n+1)(m+1)} > 2\xi$ .

Figure 1: Repartition of the mass for matrices $\mathbf{\Gamma}$ and $\mathbf{T}$. Each of them has a total mass of $\|\mathbf{q}\|_1 + \|\mathbf{p}\|_1 - s$. From a matrix $\mathbf{T}$, one can build a coupling matrix $\mathbf{\Gamma}$ by arbitrary moving some mass $T_{ij} > 0$ towards the marginal, until we get $\sum_{i=1}^{n}\sum_{j=1}^{m}\Gamma_{ij} = s$.

### 1.2.4   We then prove Proposition 1

Let us denote $\bar{\mathbf{T}}^{\backslash *}$ the matrix $\bar{\mathbf{T}}^{*}$ deprived from its last row and column. We then show that $\bar{\mathbf{T}}^{\backslash *} = \mathbf{T}^{*}$ where $\mathbf{T}^{*}$ is the solution of the original partial-W problem:

$$\mathbf{T}^{*} = \operatorname*{argmin}_{\mathbf{T} \in \Pi^{u}(\mathbf{p}, \mathbf{q})} \sum_{i=1}^{n}\sum_{j=1}^{m}C_{ij}T_{ij}.$$

We have

$$\bar{\boldsymbol{T}}^* = \operatorname*{argmin}_{\bar{\boldsymbol{T}} \in \Pi(\bar{\boldsymbol{p}}, \bar{\boldsymbol{q}})} \sum_{i=1}^{n+1} \sum_{j=1}^{m+1} \bar{C}_{ij} \bar{T}_{ij}$$

$$= \operatorname*{argmin}_{\bar{\boldsymbol{T}} \in \Pi(\bar{\boldsymbol{p}}, \bar{\boldsymbol{q}})} \sum_{i=1}^{n} \sum_{j=1}^{m} C_{ij} \bar{T}_{ij} + \xi(\|\boldsymbol{p}\|_1 + \|\boldsymbol{q}\|_1 - 2s)$$

$$= \operatorname*{argmin}_{\bar{\boldsymbol{T}} \in \Pi(\bar{\boldsymbol{p}}, \bar{\boldsymbol{q}})} \sum_{i=1}^{n} \sum_{j=1}^{m} C_{ij} \bar{T}_{ij} + \text{constant}$$

As we have $\bar{\boldsymbol{p}} = [\boldsymbol{p}, \|\boldsymbol{q}\|_1 - s]$ and $\bar{\boldsymbol{q}} = [\boldsymbol{q}, \|\boldsymbol{p}\|_1 - s]$, we can write

$$\sum_{i=1}^{n+1} T_{ij} = q_j, \forall j \le m \implies \sum_{i=1}^{n} T_{ij} \le q_j \implies (\bar{\boldsymbol{T}}^{\backslash *})^\top \mathbb{1}_n \le \boldsymbol{q}$$

$$\sum_{j=1}^{m+1} T_{ij} = p_i, \forall i \le n \implies \sum_{j=1}^{m} T_{ij} \le p_i \implies \bar{\boldsymbol{T}}^{\backslash *} \mathbb{1}_m \le \boldsymbol{p}$$

We also have $\sum_{i=1}^{n} \sum_{j=1}^{m} \bar{T}_{ij}^{\backslash *} = s$. Finally, we can write that $\bar{\boldsymbol{T}}^{\backslash *}$ belongs to the following constraint set:

$$\{\bar{\boldsymbol{T}}^{\backslash *} \in \mathbb{R}_+^{n \times m} | \bar{\boldsymbol{T}}^{\backslash *} \mathbb{1}_m \le \boldsymbol{p}, (\bar{\boldsymbol{T}}^{\backslash *})^\top \mathbb{1}_n \le \boldsymbol{q}, \mathbb{1}_n^\top \bar{\boldsymbol{T}}^{\backslash *} \mathbb{1}_m = s\}$$

which is the same like $\Pi^u(\boldsymbol{p}, \boldsymbol{q})$. We then reach the result

$$\bar{\boldsymbol{T}}^{\backslash *} = \operatorname*{argmin}_{\boldsymbol{T} \in \Pi^u(\boldsymbol{p}, \boldsymbol{q})} \sum_{i=1}^{n} \sum_{j=1}^{m} C_{ij} T_{ij} = \boldsymbol{T}^*$$

Finally we can write:

$$W_p^p(\bar{\boldsymbol{p}}, \bar{\boldsymbol{q}}) - PW_p^p(\boldsymbol{p}, \boldsymbol{q}) = \xi(\|\boldsymbol{p}\|_1 + \|\boldsymbol{q}\|_1 - 2s).$$

as long as $\bar{C}_{(n+1)(m+1)} = 2\xi + A$, with $A > 0$ and

$$W_p^p(\bar{\boldsymbol{p}}, \bar{\boldsymbol{q}}) = PW_p^p(\boldsymbol{p}, \boldsymbol{q})$$

when $\xi = 0$.

### 1.2.5 We finally show how to construct $\bar{\boldsymbol{T}}^*$ from $\boldsymbol{T}^*$

We have

$$\boldsymbol{T}^* = \operatorname*{argmin}_{\boldsymbol{T} \in \Pi^u(\boldsymbol{p}, \boldsymbol{q})} \sum_{i=1}^{n} \sum_{j=1}^{m} C_{ij} T_{ij} = \operatorname*{argmin}_{\boldsymbol{T} \in \Pi^u(\boldsymbol{p}, \boldsymbol{q})} \sum_{i=1}^{n} \sum_{j=1}^{m} \bar{C}_{ij} T_{ij} = \bar{\boldsymbol{T}}^{\backslash *}$$

Setting $\bar{T}_{(n+1)(m+1)}^* = 0$, $\bar{T}_{i(m+1)}^* = \bar{p}_i - \sum_{j=1}^{m} \bar{T}_{ij}^*$ and $\bar{T}_{(n+1)j}^* = \bar{q}_j - \sum_{i=1}^{n} \bar{T}_{ij}^*$, we recover the constraint set $\Pi(\bar{\boldsymbol{p}}, \bar{\boldsymbol{q}})$ and finally $\bar{\boldsymbol{T}}^*$.

## 1.3 Adding dummy points to the GW problem does not solve a partial-GW problem

While solving the partial-W problem can be achieved by adding dummy points and extending the cost matrix $\boldsymbol{C}$ in an appropriate way, the same strategy can not be set up for solving GW. Indeed, the equivalence between the partial and the extended problem is permitted because we can ensure that (as long as $A > 0$ and $\xi$ is a positive scalar) $\bar{T}_{(n+1)(m+1)}^* = 0$, which implies that $\sum_{i=1}^{n} \sum_{j=1}^{m} \bar{T}_{ij}^* = s$. If we extend the intra-domain cost matrices $\boldsymbol{C}^s$ and $\boldsymbol{C}^t$ on the same pattern as follows

$$\bar{\boldsymbol{C}}^s = \begin{bmatrix} \boldsymbol{C}^s & \xi \mathbb{1}_n \\ \xi \mathbb{1}_n^\top & A \end{bmatrix} \text{ and } \bar{\boldsymbol{C}}^t = \begin{bmatrix} \boldsymbol{C}^t & \xi \mathbb{1}_m \\ \xi \mathbb{1}_m^\top & A \end{bmatrix}$$

(with a constant $A > 2\xi$) the GW formulation involves pairs of points. We have

$$\bar{\boldsymbol{T}}^* = \operatorname*{argmin}_{\boldsymbol{T} \in \Pi(\bar{\boldsymbol{p}}, \bar{\boldsymbol{q}})} \sum_{i,k=1}^{n} \sum_{j,l=1}^{m} \left( C_{ik}^t - C_{jl}^s \right)^2 T_{ij} T_{kl} + (\star)$$

where

$$
\begin{aligned}
(\star) \quad = \quad & 2 \sum_{i,k=1}^{n} \left( \left( C_{ik}^t - A \right)^2 T_{i(m+1)} T_{k(m+1)} + \sum_{j=1}^{m} \left( C_{ik}^t - \xi \right)^2 T_{ij} T_{k(m+1)} \right) \\
+ \quad & 2 \sum_{j,l=1}^{m} \left( \left( A - C_{jl}^s \right)^2 T_{(n+1)j} T_{(n+1)l} + \sum_{i=1}^{n} \left( \xi - C_{jl}^s \right) T_{ij} T_{(n+1)l} \right) \\
+ \quad & 2 \sum_{j=1}^{m} \left( A - \xi \right)^2 T_{(n+1)j} T_{(n+1)(m+1)} + 2 \sum_{i=1}^{n} \left( \xi - A \right)^2 T_{i(m+1)} T_{(n+1)(m+1)} \\
+ \quad & 2 \sum_{i=1}^{n} \sum_{j=1}^{n} \left( C_{ik}^t - \xi \right)^2 T_{ij} T_{(n+1)(m+1)} + 0,
\end{aligned}
$$

does not allows having $\bar{T}^*_{(n+1)(m+1)} = 0$, and hence $\sum_{i=1}^{n} \sum_{j=1}^{m} \bar{T}^*_{ij} \neq s$.

## 2 Details of Frank-Wolfe algorithm for partial-GW

### 2.1 Line-search

The step size in the line-search of Frank-Wolfe algorithm for partial-GW is given by

$$\gamma_{\min}^{(k)} \leftarrow \operatorname*{argmin}_{\gamma \in [0,1]} \left\{ \mathcal{J}_{\boldsymbol{C}^s, \boldsymbol{C}^t} \left( (1 - \gamma) \boldsymbol{T}^{(k)} + \gamma \widetilde{\boldsymbol{T}}^{(k)} \right) \right\}.$$

Define $\boldsymbol{E}^{(k)} = \widetilde{\boldsymbol{T}}^{(k)} - \boldsymbol{T}^{(k)}$ and the function $\phi : [0,1] \to \mathbb{R}$ such that

$$\phi(\gamma^{(k)}) = \mathcal{J}_{\boldsymbol{C}^s, \boldsymbol{C}^t} (\boldsymbol{T}^{(k)} + \gamma \boldsymbol{E}^{(k)}).$$

Straightforwardly, one has

$$
\begin{aligned}
\phi(\gamma) &= \langle \mathcal{M}(\boldsymbol{C}^s, \boldsymbol{C}^t) \circ (\boldsymbol{T}^{(k)} + \gamma \boldsymbol{E}^{(k)}), \boldsymbol{T}^{(k)} + \gamma \boldsymbol{E}^{(k)} \rangle_F \\
&= \langle \mathcal{M}(\boldsymbol{C}^s, \boldsymbol{C}^t) \circ \boldsymbol{T}^{(k)}, \boldsymbol{T}^{(k)} \rangle_F + \gamma \langle \mathcal{M}(\boldsymbol{C}^s, \boldsymbol{C}^t) \circ \boldsymbol{T}^{(k)}, \boldsymbol{E}^{(k)} \rangle_F + \gamma \langle \mathcal{M}(\boldsymbol{C}^s, \boldsymbol{C}^t) \circ \boldsymbol{E}^{(k)}, \boldsymbol{T}^{(k)} \rangle_F \\
&\quad + \gamma^2 \langle \mathcal{M}(\boldsymbol{C}^s, \boldsymbol{C}^t) \circ \boldsymbol{E}^{(k)}, \boldsymbol{E}^{(k)} \rangle_F.
\end{aligned}
$$

Since we choose a quadratic cost, $p = 2$, then for any $\boldsymbol{T}, \boldsymbol{R}$ one has $\langle \mathcal{M}(\boldsymbol{C}^s, \boldsymbol{C}^t) \circ \boldsymbol{R}, \boldsymbol{T} \rangle_F = \langle \mathcal{M}(\boldsymbol{C}^s, \boldsymbol{C}^t) \circ \boldsymbol{T}, \boldsymbol{R} \rangle_F$ and we can then rewrite

$$\phi(\gamma) = \gamma^2 \langle \mathcal{M}(\boldsymbol{C}^s, \boldsymbol{C}^t) \circ \boldsymbol{E}^{(k)}, \boldsymbol{E}^{(k)} \rangle_F + 2\gamma \langle \mathcal{M}(\boldsymbol{C}^s, \boldsymbol{C}^t) \circ \boldsymbol{E}^{(k)}, \boldsymbol{T}^{(k)} \rangle_F + \langle \mathcal{M}(\boldsymbol{C}^s, \boldsymbol{C}^t) \circ \boldsymbol{T}^{(k)}, \boldsymbol{T}^{(k)} \rangle_F.$$

We then have to find $\gamma_o^{(k)}$ that minimises $\phi(\gamma) = a\gamma^2 + b\gamma + c$, with

$$a = \langle \mathcal{M}(\boldsymbol{C}^s, \boldsymbol{C}^t) \circ \boldsymbol{E}^{(k)}, \boldsymbol{E}^{(k)} \rangle_F, \quad b = 2\langle \mathcal{M}(\boldsymbol{C}^s, \boldsymbol{C}^t) \circ \boldsymbol{E}^{(k)}, \boldsymbol{T}^{(k)} \rangle_F, \quad c = \langle \mathcal{M}(\boldsymbol{C}^s, \boldsymbol{C}^t) \circ \boldsymbol{T}^{(k)}, \boldsymbol{T}^{(k)} \rangle_F$$

with its derivative $\phi'(\gamma) = 2a\gamma + b$. This yields the following cases:

**Case 1: $a > 0$.** In that case, $\phi(\gamma)$ is a convex function, whose minimum on $[0, 1]$ is reached for

$$\gamma_{\min}^{(k)} = \min\{ -\frac{b}{2a}, 1 \}.$$

Indeed, we have for $k \geq 1$

$$
\begin{aligned}
\widetilde{\boldsymbol{T}}^{(k)} &= \operatorname*{argmin}_{\boldsymbol{T} \in \Pi(\boldsymbol{p}, \boldsymbol{q})} \langle \nabla \mathcal{J}_{\boldsymbol{C}^s, \boldsymbol{C}^t} (\boldsymbol{T}^{(k)}), \boldsymbol{T} \rangle_F \\
&= \operatorname*{argmin}_{\boldsymbol{T} \in \Pi(\boldsymbol{p}, \boldsymbol{q})} \langle \mathcal{M}(\boldsymbol{C}^s, \boldsymbol{C}^t) \circ \boldsymbol{T}^{(k)}, \boldsymbol{T} \rangle_F
\end{aligned}
$$

Hence
$$\langle \mathcal{M}(\boldsymbol{C}^s, \boldsymbol{C}^t) \circ \boldsymbol{T}^{(k)}, \widetilde{\boldsymbol{T}}^{(k)} \rangle_F \leq \langle \mathcal{M}(\boldsymbol{C}^s, \boldsymbol{C}^t) \circ \boldsymbol{T}^{(k)}, \boldsymbol{T}^{(k)} \rangle_F$$
$$\langle \mathcal{M}(\boldsymbol{C}^s, \boldsymbol{C}^t) \circ \widetilde{\boldsymbol{T}}^{(k)}, \boldsymbol{T}^{(k)} \rangle_F \leq \langle \mathcal{M}(\boldsymbol{C}^s, \boldsymbol{C}^t) \circ \boldsymbol{T}^{(k)}, \boldsymbol{T}^{(k)} \rangle_F$$
$$\langle \mathcal{M}(\boldsymbol{C}^s, \boldsymbol{C}^t) \circ \boldsymbol{E}^{(k)}, \boldsymbol{T}^{(k)} \rangle_F \leq 0.$$
Then, $b = 2\langle \mathcal{M}(\boldsymbol{C}^s, \boldsymbol{C}^t) \circ \boldsymbol{E}^{(k)}, \boldsymbol{T}^{(k)} \rangle_F \leq 0$ hence $-\frac{b}{2a} \geq 0$.

**Case 2: $a < 0$.** In that case, $\phi(\gamma)$ is a concave function, whose minimum is reached either for $\gamma = 0$ or $\gamma = 1$. We have $\phi(0) = c > 0$ and $\phi(1) = a + b + c$. The minimum is then obtained for 0 if $a + b > 0$, and 1 otherwise. We have previously shown that $b \leq 0$, which implies that
$$\gamma_{\min}^{(k)} = 1.$$

## 2.2 Convergence guarantee.

Intuitively a stationary point $\boldsymbol{T}^o$ for partial-GW problem verifies that every direction in the polytope with origin $\boldsymbol{T}^o$ is correlated with the gradient of the loss $\mathcal{J}_{\boldsymbol{C}^s, \boldsymbol{C}^t}(\cdot)$, namely $\langle \nabla \mathcal{J}_{\boldsymbol{C}^s, \boldsymbol{C}^t}(\boldsymbol{T}^o); \boldsymbol{T} - \boldsymbol{T}^o \rangle_F \geq 0$ for all $\boldsymbol{T} \in \Pi^u(\boldsymbol{p}, \boldsymbol{q})$. A good criterion to measure distance to a stationary point at iteration $k$ is the often used *Frank-Wolfe gap*, which is defined by
$$g_k = \langle \nabla \mathcal{J}_{\boldsymbol{C}^s, \boldsymbol{C}^t}(\boldsymbol{T}^{(k)}), \boldsymbol{T}^{(k)} - \widetilde{\boldsymbol{T}}^{(k)} \rangle_F.$$
Note that $g_k$ is always non-negative, and zero if and only if at a stationary point. Thanks to Theorem 1 in Lacoste-Julien (2016) we have
$$\min_{0 \leq k \leq K} g_k \leq \frac{\max(2J_0, L\mathrm{diam}(\Pi^u(\boldsymbol{p}, \boldsymbol{q}))^2)}{\sqrt{K+1}},$$
where $J_0 = \mathcal{J}_{\boldsymbol{C}^s, \boldsymbol{C}^t}(\boldsymbol{T}^{(0)}) - \min_{\boldsymbol{T} \in \Pi^u(\boldsymbol{p}, \boldsymbol{q})} \mathcal{J}_{\boldsymbol{C}^s, \boldsymbol{C}^t}(\boldsymbol{T})$ defines the initial suboptimality, $L$ is a Lipschitz constant of $\nabla \mathcal{J}_{\boldsymbol{C}^s, \boldsymbol{C}^t}$ and $\mathrm{diam}(\Pi^u(\boldsymbol{p}, \boldsymbol{q}))$ is the $\|\cdot\|_F$-diameter of $\Pi^u(\boldsymbol{p}, \boldsymbol{q})$ (see (4)). Therefore we can state the following lemma characterizing the convergence guarantee:

**Lemma 1** *The Frank-Wolfe gap $g_K = \min_{0 \leq k \leq K} g_k$ for the partial-GW loss $\mathcal{J}_{\boldsymbol{C}^s, \boldsymbol{C}^t}$ after $K$ iterations satisfies*
$$g_K \leq \frac{2\max(J_0, \sqrt{2}s(\max(C_{ij}^s) + \max(C_{ij}^t)))}{\sqrt{K+1}} \tag{3}$$
*where $s$ is the total mass to be transported and $\max(C_{ij}^s)$ is the maximum value of cost matrix $\max(C_{ij}^t)$, similarly for $\boldsymbol{C}^t$.*

Note that for the implementations, one can set $\max(C_{ij}^s) = 1 = \max(C_{ij}^t)$, hence the upper bound in (3) becomes more tight regarding a good initialization of Algorithm 1. This can be used to reduce significantly the initial suboptimality $J_0$. Furthermore, according to Theorem 1 in Lacoste-Julien (2016), Algorithm 1 takes at most $\mathcal{O}(1/\varepsilon^2)$ iterations to find an approximate stationary point with a gap smaller than $\varepsilon$.

*Proof of Lemma 1.* Let us first calculate the diameter of the couplings set $\Pi^u(\boldsymbol{p}, \boldsymbol{q})$ with respect to the Frobenieus norm $\|\cdot\|_F$. One has
$$\mathrm{diam}(\Pi^u(\boldsymbol{p}, \boldsymbol{q})) = \sup_{(\boldsymbol{T}, \boldsymbol{Q}) \in \Pi^u(\boldsymbol{p}, \boldsymbol{q})^2} \|\boldsymbol{T} - \boldsymbol{Q}\|_F. \tag{4}$$
Using triangle inequality and the fact that $T_{ij}, Q_{ij}$ are probability masses that is $T_{ij}, Q_{ij} \in [0, 1]$, we get
$$\|\boldsymbol{T} - \boldsymbol{Q}\|_F^2 \leq 2\|\boldsymbol{T}\|_F^2 + 2\|\boldsymbol{Q}\|_F^2$$
$$\leq 2\sum_{i,j}^{n,m} T_{ij}^2 + 2\sum_{i,j}^{n,m} Q_{ij}^2$$
$$\leq 2\sum_{i,j}^{n,m} T_{ij} + 2\sum_{i,j}^{n,m} Q_{ij}$$
$$\leq 4s$$

where $s$ in the total mass to be transported. Thus $\mathrm{diam}(\Pi^u(\boldsymbol{p}, \boldsymbol{q})) \leq 2\sqrt{s}$.

For the Lipschitz constant of the gradient of $\mathcal{J}_{\boldsymbol{C}^s, \boldsymbol{C}^t}$ we proceed as follows: for any $\boldsymbol{T}, \boldsymbol{Q} \in \Pi^u(\boldsymbol{p}, \boldsymbol{q})$ we have

$$\|\nabla\mathcal{J}_{\boldsymbol{C}^s, \boldsymbol{C}^t}(\boldsymbol{T}) - \nabla\mathcal{J}_{\boldsymbol{C}^s, \boldsymbol{C}^t}(\boldsymbol{Q})\|_F^2 = \sum_{i,j}^{n,m} \Big(\sum_{k,l}^{n,m} \mathcal{M}_{ijkl}(T_{kl} - Q_{kl})\Big)^2$$

$$\leq \sup_{i,j,k,l} \mathcal{M}_{ijkl}^2 \sum_{i,j}^{n,m} \Big(\sum_{k,l}^{n,m}(T_{kl} - Q_{kl})\Big)^2$$

$$\leq 4 \sup_{i,j,k,l} \mathcal{M}_{ijkl}^2 \|\boldsymbol{T} - \boldsymbol{Q}\|_F^2.$$

We have also

$$\sup_{i,j,k,l} \mathcal{M}_{ijkl}^2 = \frac{1}{4} \sup_{i,j,k,l} (C_{ik}^s - C_{jl})^2 \leq \frac{1}{2}((\max(C_{ij}^s))^2 + (\max(C_{kl}^t))^2).$$

Hence the Lipschitz constant of $\nabla\mathcal{J}_{\boldsymbol{C}^s, \boldsymbol{C}^t}(\cdot)$ verifies $L \leq \sqrt{2}(\max(C_{ij}^s) + \max(C_{kl}^t))$. This gives the desired result.

# 3 Equivalence between the regularized extended problem and the PU learning problem and proof of Proposition 2

Let's denote $\bar{\boldsymbol{T}}^*$ the optimal coupling of the extended problem

$$\bar{\boldsymbol{T}}^* = \underset{\bar{\boldsymbol{T}} \in \Pi(\bar{\boldsymbol{p}}, \bar{\boldsymbol{q}})}{\mathrm{argmin}} \sum_{i=1}^{n+1} \sum_{j=1}^{m+1} \bar{C}_{ij} \bar{T}_{ij} + \eta\Omega(\bar{\boldsymbol{T}}) \tag{5}$$

in which $\Omega(\bar{\boldsymbol{T}}) = \sum_{i=1}^{n} \left(\|\bar{\boldsymbol{T}}_{i(:m)}\|_2 + \|\bar{\boldsymbol{T}}_{i(m+1)}\|_2\right)$ where $\bar{\boldsymbol{T}}_{i(:m)}$ is a vector that contains the values of the $i^{\text{th}}$ line of $\bar{\boldsymbol{T}}$ associated to the first $m$ columns.

## 3.1 We first show that $\sum_{i=1}^{n} \sum_{j=1}^{m} \bar{T}_{ij}^* = s$

Let recall that by construction, we have $\bar{\boldsymbol{p}} = [\boldsymbol{p}, \quad \|\boldsymbol{q}\|_1 - s]$ and $\bar{\boldsymbol{q}} = [\boldsymbol{q}, \quad \|\boldsymbol{p}\|_1 - s]$. Also due to the definition of the marginals $p_i = \frac{1-\alpha}{n}$ for $i = 1, \cdots, n$ and $q_j = \frac{s+\alpha}{m}$ for $j = 1, \cdots, m$ we get $\|\boldsymbol{p}\|_1 = 1 - \alpha$, $\|\boldsymbol{q}\|_1 = s + \alpha$.

Therefore we arrive at the results $\sum_{i=1}^{n+1} \sum_{j=1}^{m+1} \bar{T}_{ij}^* = 1$ by construction. Using the development in Section 1.2.2 of the supplemental, we can establish that $\sum_{i=1}^{n+1} \bar{T}_{i(m+1)}^* = 1 - \alpha - s$, and $\sum_{j=1}^{m+1} \bar{T}_{(n+1)j}^* = \alpha$. Thereon we attain $\sum_{i=1}^{n} \sum_{j=1}^{m} \bar{T}_{ij}^* = 1 - (1 - \alpha - s + \alpha) = s$.

## 3.2 Proof of Proposition 2

Given a solution $\bar{\boldsymbol{T}}^*$ of the extended PU learning problem stated in Equation (5) of the supplementary, we can write the objective function of the extended problem as

$$\sum_{i=1}^{n+1} \sum_{j=1}^{m+1} \bar{C}_{ij} \bar{T}_{ij}^* = \sum_{i=1}^{n} \sum_{j=1}^{m} C_{ij} \bar{T}_{ij}^* + \xi \sum_{i=1}^{n} \bar{T}_{i(m+1)}^* + \xi \sum_{j=1}^{m} \bar{T}_{(n+1)j}^* + (2\xi + A)\bar{T}_{(n+1)(m+1)}^*$$

$$= \sum_{i=1}^{n} \sum_{j=1}^{m} C_{ij} \bar{T}_{ij}^* + \xi(1 - \alpha - s + \alpha) + 0$$

$$= \sum_{i=1}^{n} \sum_{j=1}^{m} C_{ij} \bar{T}_{ij}^* + \xi(1 - s).$$

Let's then show that $(\bar{\boldsymbol{T}}^*)_{i,j=1}^{n,m} = \boldsymbol{T}^*$, where $\boldsymbol{T}^*$ is the solution of the original PU problem:

$$\boldsymbol{T}^* = \operatorname*{argmin}_{\boldsymbol{T}\in\Pi^{PU}(\boldsymbol{p},\boldsymbol{q})} \sum_{i=1}^{n}\sum_{j=1}^{m} C_{ij}T_{ij}.$$

As shown before, we know that $\bar{T}^*_{(n+1)(m+1)} = 0$, hence we can rewrite the extended problem as

$$\bar{\boldsymbol{T}}^* = \operatorname*{argmin}_{\bar{\boldsymbol{T}}\in\Pi(\bar{\boldsymbol{p}},\bar{\boldsymbol{q}})} \sum_{i=1}^{n}\sum_{j=1}^{m} C_{ij}\bar{T}_{ij} + \eta \sum_{i=1}^{n} \left( \|\bar{\boldsymbol{T}}_{i(:m)}\|_2 + \|\bar{\boldsymbol{T}}_{i(m+1)}\|_2 \right).$$

We now turn this problem into its equivalent constrained form, namely there exists some $\lambda > 0$, related to the regularization parameter $\eta$, such that

$$\bar{\boldsymbol{T}}^* = \begin{cases} \operatorname{argmin}_{\bar{\boldsymbol{T}}\in\Pi(\bar{\boldsymbol{p}},\bar{\boldsymbol{q}})} \sum_{i=1}^{n}\sum_{j=1}^{m} C_{ij}\bar{T}_{ij} \\ \text{s.t. } \sum_{i=1}^{n} \|\bar{\boldsymbol{T}}_{i(:m)}\|_2 + \|\bar{\boldsymbol{T}}_{i(m+1)}\|_2 \leq \lambda \end{cases}$$

Let $\bar{\boldsymbol{T}}^{\backslash *} \in \mathbb{R}^{n\times m}$ such that $\bar{T}_{ij}^{\backslash *} = \bar{T}_{ij}$ for all $i,j \in \{1,\ldots,n\}\times\{1,\ldots,m\}$. Since we have the polytope constraint $\sum_{j=1}^{m+1}\bar{T}_{ij} = p_i$, $\forall i$, this means that $\sum_{j=1}^{m}\bar{T}_{ij} \leq p_i$ and analagously $\sum_{i=1}^{n}\bar{T}_{ij} \leq q_j$. Using the established results from section 3.1 of the supplementary, we can derive that the matrix $\bar{\boldsymbol{T}}^{\backslash *} \in \mathbb{R}^{n\times m}$ belongs to the following constraint set:

$$\Pi^u(\boldsymbol{p},\boldsymbol{q}) = \{\boldsymbol{Q}\in\mathbb{R}_+^{n\times m} | \boldsymbol{Q}\mathbb{1}_m \leq \boldsymbol{p}, \boldsymbol{Q}^\top\mathbb{1}_n \leq \boldsymbol{q}, \mathbb{1}_n^\top\boldsymbol{Q}\mathbb{1}_m = s\}.$$

Let define $\mathbf{u}_i = \bar{T}_{i(:m)}$ for all $i = 1,\ldots,n$ and $\mathbf{v} = \bar{T}_{(:n)(m+1)} \in \mathbb{R}^n$. Then the problem can be re-formulated as

$$\bar{\boldsymbol{T}}^* = \begin{cases} \operatorname*{argmin}_{\bar{\boldsymbol{T}}^{\backslash *}\in\Pi^u(\boldsymbol{p},\boldsymbol{q}),\mathbf{u}_i,\mathbf{v},\mathbf{w}} \sum_{i=1}^{n}\sum_{j=1}^{m} C_{ij}\bar{T}_{ij}^{\backslash *} \\ \text{s.t. } \sum_{i=1}^{n} \|\mathbf{u}_i\|_2 + \|\mathbf{v}\|_1 \leq \lambda \end{cases}$$

On the other hand, using the polytope constraints we have

$$\sum_{i=1}^{n}\|\mathbf{u}_i\|_2 = \sum_{i=1}^{n}\sqrt{\sum_{j=1}^{m}\bar{T}_{ij}^2} \leq \sum_{i=1}^{n}\sqrt{\sum_{j=1}^{m}\bar{T}_{ij}}$$

$$\leq \sum_{i=1}^{n}\sqrt{p_i} = \sum_{i=1}^{n}\sqrt{\frac{1-\alpha}{n}} = \sqrt{n}\sqrt{1-\alpha}.$$

and

$$\|\mathbf{v}\|_1 = \sum_{i=1}^{n}\bar{T}_{i(m+1)} = q_{m+1} = 1 - \alpha - s.$$

Gathering those results, we get

$$\sum_{i=1}^{n}\|\mathbf{u}_i\|_2 + \|\mathbf{v}\|_1 \leq \sqrt{n}\sqrt{1-\alpha} + (1-\alpha-s).$$

Therefore, for any value of $\lambda$ such that $\lambda > \lambda_b := \sqrt{n}\sqrt{1-\alpha}+(1-\alpha-s)$, the group-lasso constraint $\sum_{i=1}^{n}\|\mathbf{u}_i\|_2 + \|\mathbf{v}\|_1 \leq \lambda$ is always satisfied for any choice of $\mathbf{u}_i, \mathbf{v}$ verifying $\|\mathbf{v}\|_1 = 1 - \alpha - s$ and the marginal constraints

$$p_i = \|\mathbf{u}_i\|_1 + \bar{T}_{i(m+1)}, \qquad \forall i. \tag{6}$$

One can choose a particular sparse solution for $\mathbf{u}_i$ as follows:

- if $\bar{T}_{i(m+1)} = 0$, for $i \in \{1,\ldots,n\}$, condition (6) implies necessary that $\sum_{j=1}^{m}\bar{T}_{ij}^{\backslash *} = \sum_{j=1}^{m}\bar{T}_{ij} = \|\mathbf{u}_i\|_1 = p_i$.

- If not, one can choose $\mathbf{u}_i$ such that $\|\mathbf{u}_i\|_1 = 0$, and hence $\sum_{j=1}^{m}\bar{T}_{ij} = \sum_{j=1}^{m}\bar{T}_{ij}^{\backslash *} = 0$.

So it remains that the solution of the constrained problem is such that

$$\bar{\boldsymbol{T}}^{\backslash *} \in \Pi^{PU}(\boldsymbol{p}, \boldsymbol{q}) = \{\boldsymbol{Q} \in \mathbb{R}_+^{n \times m} | \boldsymbol{Q} \mathbb{1}_m = \{\boldsymbol{p}, 0\}, \boldsymbol{Q}^\top \mathbb{1}_n \leq \boldsymbol{q}, \mathbb{1}_n^\top \boldsymbol{Q} \mathbb{1}_m = s\},$$

that is either $\sum_{j=1}^m \bar{T}_{ij}$ or $\bar{T}_{i(m+1)}$ is exactly 0. This means that there exists some value $\eta_b$, related to $\lambda_b = \sqrt{n}\sqrt{1-\alpha} + (1-\alpha-s)$, such that for $\eta \geq \eta_b$, solving the extended problem (5) amounts to solving our PU learning formulation, which concludes the proof.

## 4    Initialization of partial-W and -GW

The partial-OT computation is based on a augmented problem with a dummy point and, as such, is convex. On the contrary, the GW problem is non-convex and, although the algorithm is proved to converge, there is no guarantee that the global optimum is reached. The quality of the solution is therefore highly dependent on the initialization. We propose to rely on an initial Wasserstein barycenter problem to build a first guess of the transport matrix.

For partial-GW, as the $\boldsymbol{C}^s$ and $\boldsymbol{C}^t$ matrices do not lie in the same ground space, we can not define a distance function between their members. Nevertheless, within a domain, we can build two sets of "homogeneous" points. Instead of relying on a classical partitioning algorithm such as $k$-means, we propose to look for a barycenter with atoms $\mathbf{U}_2 = [\boldsymbol{u}_2^1, \boldsymbol{u}_2^1]$ and weights $\boldsymbol{b}$ of the set $\mathbf{U}$ that minimizes the following function:

$$f(\boldsymbol{b}, \mathbf{U}_2) = W_p^p(\boldsymbol{b}, \boldsymbol{q}) \tag{7}$$

over the feasible sets for $\mathbf{U}_2$, where $\boldsymbol{b} = [\pi, 1-\pi]$. In other word, we look for the set $\mathbf{U}_2$ that allows having the most similar (in the Wasserstein sense) distribution $\boldsymbol{q}$ as $\mathbf{U}$. The induced transport matrix gives then two clusters, the one with mass $\pi$ serving as an initialization matrix for the GW problem.

Whenever possible (that is to say when **Pos** and **Unl** belong to the same space), we also initialize the GW algorithm with the solution of Partial-W, and the outer product of $\bar{\boldsymbol{p}}$ and $\bar{\boldsymbol{q}}$.

## 5    Effect of the group constraints on the Wasserstein coupling.

We first draw $n_P = 10$ and $n_u = 10$ samples, 6 of them being positives and we set $\alpha = 0$. Fig. 5 shows the data and the obtained optimal couplings: enforcing a group constraint assigns some unlabeled points to the dummy point, allowing a clear identification of the negatives among **Unl**, whereas the solution with no such constraints may split the probability mass of the unlabeled positives between **Pos** and the dummy point, preventing to consistently identify the negatives.

Figure 2: (Left) Positives (black) and Unlabeled (blue) samples (Middle) Transport matrix with no group constraints, where darker color indicates stronger matching (Right) Transport matrix with group constraints."D" is the dummy point.

Table 1: Standard deviation of accuracy rates on different datasets and scenarii.

| DATASET/SCENARIO | $\pi$ | PU | PUSB | P-W 0 | P-W 0.025 | P-GW 0 | P-W 0.025 |
|---|---|---|---|---|---|---|---|
| MUSHROOMS | 51.8 | 0.040 | 0.047 | 0.012 | 0.008 | 0.008 | 0.011 |
| SHUTTLE | 78.6 | 0.032 | 0.045 | 0.009 | 0.012 | 0.018 | 0.017 |
| PAGEBLOCKS | 89.8 | 0.012 | 0.010 | 0.014 | 0.007 | 0.012 | 0.010 |
| USPS | 16.7 | 0.012 | 0.014 | 0.004 | 0.006 | 0.020 | 0.021 |
| CONNECT-4 | 65.8 | 0.003 | 0.010 | 0.017 | 0.016 | 0.019 | 0.017 |
| SPAMBASE | 39.4 | 0.036 | 0.039 | 0.026 | 0.023 | 0.022 | 0.021 |
| ORIGINAL MNIST | 10 | 0.005 | 0.005 | 0.005 | 0.004 | 0.004 | 0.01 |
| COLORED MNIST | 10 | 0.035 | 0.034 | 0.004 | 0.006 | 0.008 | 0.012 |
| SURF C→SURF C | 10 | 0.019 | 0.015 | 0.017 | 0.014 | 0.015 | 0.017 |
| SURF C→SURF A | 10 | 0.014 | 0.018 | 0.012 | 0.014 | 0.018 | 0.016 |
| SURF C→SURF W | 10 | 0.010 | 0.064 | 0.014 | 0.013 | 0.016 | 0.010 |
| SURF C→SURF D | 10 | 0.006 | 0.036 | 0.000 | 0.000 | 0.022 | 0.020 |
| DECAF C→DECAF C | 10 | 0.022 | 0.017 | 0.013 | 0.016 | 0.013 | 0.009 |
| DECAF C→DECAF A | 10 | 0.019 | 0.022 | 0.006 | 0.006 | 0.004 | 0.014 |
| DECAF C→DECAF W | 10 | 0.030 | 0.009 | 0.006 | 0.006 | 0.014 |  |
| DECAF C→DECAF D | 10 | 0.033 | 0.008 | 0.010 | 0.009 | 0.015 | 0.003 |
| SURF C→DECAF C | 10 | - | - | - | - | 0.012 | 0.013 |
| SURF C→DECAF A | 10 | - | - | - | - | 0.011 | 0.011 |
| SURF C→DECAF W | 10 | - | - | - | - | 0.006 | 006 |
| SURF C→DECAF D | 10 | - | - | - | - | 0.010 | 0.010 |
| DECAF C→SURF C | 10 | - | - | - | - | 0.024 | 0.024 |
| DECAF C→SURF A | 10 | - | - | - | - | 0.037 | 0.037 |
| DECAF C→SURF W | 10 | - | - | - | - | 0.022 | 0.022 |
| DECAF C→SURF D | 10 | - | - | - | - | 0.013 | 0.013 |

# 6 Additionnal results

## 6.1 Standard deviation of accuracy rates of the experiments

## 6.2 Timings

We report the timings related to P-W 0 and P-GW 0 (those related to P-W 0.025 and P-GW 0.025 are similar) for 1 run. As P-GW iterates over a P-W resolution, one can infer that few iterations are needed for P-GW to converge.

Table 2: Timings (in seconds) of 1 run of P-W 0 and P-GW 0.

|  | MUSH. | SHUTTLE | PAGE. | USPS | CONN. | SPAM. | OR. MNIST | COL. MNIST |
|---|---|---|---|---|---|---|---|---|
| P-W 0 | 1.4 | 0.9 | 1.2 | 1.1 | 1.4 | 1.0 | 1.4 | 1.0 |
| P-GW 0 | 2.1 | 1.9 | 5.0 | 3.9 | 3.2 | 1.8 | 5.6 | 3.6 |
|  | SURF C→ SURF * | | | | DECAF C→ DECAF * | | | |
|  | * = C | * = A | * = W | * = D | * = C | * = A | * = W | * = D |
| P-W 0 | 0.1 | 0.1 | 0.1 | 0.1 | 0.1 | 0.1 | 0.1 | 0.1 |
| P-GW 0 | 0.2 | 0.2 | 0.2 | 0.3 | 0.4 | 0.6 | 0.3 | 0.3 |
|  | SURF C→ DECAF * | | | | DECAF C→ SURF * | | | |
|  | * = C | * = A | * = W | * = D | * = C | * = A | * = W | * = D |
| P-GW 0 | 0.2 | 0.2 | 0.2 | 0.2 | 0.6 | 0.3 | 0.2 | 0.5 |

## 6.3 Sensitivity to the proportion of positives w.r.t. the actual class prior

We vary the proportion of positives in the unlabeled dataset, using $s = \pi' = [0.8\pi, 0.9\pi, \cdots, 1.2\pi]$, as it is done in Kiryo et al. (2017), and report the results for P-W 0 and P-GW 0.

Table 3: Accuracy rates of P-W 0 and P-GW 0 for different datasets and proportion of positives in the unlabeled dataset.

| DATASET/SCENARIO | P-W 0 | | | | | P-GW 0 | | | | |
|---|---|---|---|---|---|---|---|---|---|---|
| | $0.8\pi$ | $0.9\pi$ | $\pi$ | $1.1\pi$ | $1.2\pi$ | $0.8\pi$ | $0.9\pi$ | $\pi$ | $1.1\pi$ | $1.2\pi$ |
| MUSHROOMS | 89.1 | 93.7 | 96.3 | 93.7 | 89.0 | 88.9 | 93.0 | 95.0 | 93.2 | 88.5 |
| SHUTTLE | 83.8 | 90.6 | 95.8 | 91.3 | 84.0 | 83.1 | 90.0 | 94.2 | 89.7 | 82.9 |
| PAGEBLOCKS | 78.2 | 86.2 | 92.2 | 80.7 | - | 81.2 | 88.5 | 90.9 | 89.8 | - |
| USPS | 96.8 | 98.1 | 98.3 | 97.5 | 95.9 | 91.7 | 93.0 | 94.9 | 94.8 | 94.1 |
| CONNECT-4 | 52.9 | 52.9 | 55.6 | 59.9 | 61.5 | 57.5 | 58.8 | 59.5 | 61.0 | 62.2 |
| SPAMBASE | 77.0 | 77.9 | 78.0 | 77.8 | 76.9 | 71.9 | 71.3 | 70.2 | 71.2 | 69.4 |
| ORIGINAL MNIST | 97.9 | 98.8 | 98.8 | 98.2 | 97.4 | 97.7 | 98.1 | 98.2 | 97.7 | 96.9 |
| COLORED MNIST | 92.0 | 91.8 | 91.5 | 91.3 | 90.9 | 96.6 | 97.3 | 97.3 | 97.5 | 97.0 |
| SURF C→SURF C | 91.2 | 90.6 | 90.0 | 89.8 | 89.2 | 87.8 | 87.0 | 87.2 | 86.4 | 86.0 |
| SURF C→SURF A | 83.6 | 82.8 | 81.6 | 80.6 | 79.8 | 87.4 | 86.4 | 85.6 | 85.0 | 83.6 |
| SURF C→SURF W | 83.6 | 82.8 | 81.6 | 81.4 | 80.4 | 86.2 | 85.8 | 85.6 | 84.6 | 84.0 |
| SURF C→SURF D | 82.0 | 81.0 | 80.0 | 79.0 | 78.0 | 88.4 | 88.6 | 87.6 | 87.0 | 86.0 |
| DECAF C→DECAF C | 93.6 | 93.4 | 94.0 | 94.0 | 94.0 | 87.8 | 87.2 | 86.4 | 86.4 | 85.8 |
| DECAF C→DECAF A | 82.2 | 81.2 | 80.2 | 79.2 | 78.2 | 90.4 | 90.4 | 89.2 | 88.8 | 88.4 |
| DECAF C→DECAF W | 82.0 | 81.0 | 80.2 | 79.2 | 78.2 | 89.4 | 89.2 | 89.2 | 88.4 | 87.6 |
| DECAF C→DECAF D | 82.0 | 81.2 | 80.8 | 80.2 | 79.0 | 93.4 | 94.2 | 94.2 | 93.4 | 93.0 |
| SURF C→DECAF C | - | - | - | - | - | 88.4 | 88.2 | 87.0 | 86.2 | 85.4 |
| SURF C→DECAF A | - | - | - | - | - | 95.2 | 94.4 | 94.4 | 93.0 | 91.4 |
| SURF C→DECAF W | - | - | - | - | - | 96.4 | 95.2 | 94.4 | 93.4 | 92.2 |
| SURF C→DECAF D | - | - | - | - | - | 97.6 | 97.6 | 97.4 | 96.6 | 95.6 |
| DECAF C→SURF C | - | - | - | - | - | 86.2 | 86.2 | 85.0 | 84.2 | 83.2 |
| DECAF C→SURF A | - | - | - | - | - | 84.8 | 84.0 | 82.6 | 82.6 | 81.8 |
| DECAF C→SURF W | - | - | - | - | - | 85.6 | 85.6 | 83.8 | 83.0 | 82.2 |
| DECAF C→SURF D | - | - | - | - | - | 84.4 | 83.2 | 82.8 | 81.0 | 80.6 |