[Reviews · NeurIPS 2020]

Review 1

Summary and Contributions: The paper found that the solution of the partial Wasserstein problem's solution could be got by solving the original extended Wasserstein problem first then dropping the extended row and column. Then the paper applies the same trick to the partial Gromov-Wasserstein problem and provides a way to solve this problem.

Strengths: Introducing the outlier bin to the optimal transport problem is not a new idea, for example in paper[1]. In this paper, the authors use this idea to solve the partial assignment problem and extend it to the partial GW problem with theoretical proof. This part is novel and the paper provides detailed mathematical proof. The proposed method has been tested on the dataset and compared with other baseline methods. Sometimes this method shows better performance. [1] SuperGlue: Learning Feature Matching with Graph Neural Networks, CVPR2020

Weaknesses: 1. In this work, although the partial assignment problem is addressed, how to set the mass of the dummy point and the kesi in the cost matrix might be a potential issue. These parameters associated with the dataset which could highly affect the result. It might be difficult to tune. 2. I'm a little bit curious about the running time, which is totally missing in the paper.

Correctness: The propose method seems like mathmatically sound and feasible in practice.

Clarity: The paper is well written.

Relation to Prior Work: The paper clearly explains the contribution.

Reproducibility: Yes

Additional Feedback: I agree the R3's opinion which the paper is difficult to read, with more explanation sentences will make the paper easier to read. About the rebuttal, it addresses all my concerns, as a result I keep my rate about this paper.


Review 2

Summary and Contributions: This paper tackles the problem of computing partial optimal transport metrics. More precisely, the authors consider both the partial-Wassserstein (partial-W) and the partial-Gromov-Wasserstein (partial-GW) distances. Their first contribution is to show that the partial-W can be reduced to a (full) Wasserstein problem by adding dummy points. For the partial-GW approach, a Frank-Wolfe algorithm is proposed, where a partial-W distance is computed at each iteration. The second contribution is to show that the PU-learning problem can be solved by using partial-W and partial-GW distances. Experiments on various datasets illustrate the efficiency of this approach.

Strengths: - This work has pedagogical qualities: the different variants of the Wasserstein distance are recalled very clearly. - The "dummy point" trick to compute the partial-W is very smart and interesting.

Weaknesses: The definition of the Gromov-Wasserstein distance should be discussed a bit more, as it is less usual than the Wasserstein distance. Otherwise it seems a bit arbitrary ... For instance, is there a case where both W and GW coincide?

Correctness: Yes, the experiments illustrate the theory.

Clarity: Yes, the paper is well written.

Relation to Prior Work: Previous related works on W and GW distances are recalled, as well as existing PU-learning methods.

Reproducibility: Yes

Additional Feedback: Typos and comments: - line 97: typo in the expression of PW - line 163: formula of the posterior probability should be p(y=1|x) - I felt a bit lost when reading the section 3: I think the use of partial-W to compute partial-GW should be stated more explicitly in the beginning of section 3 --> the authors will clarify this point - notation in section 4.2: it was a bit disturbing to me that "p" does not refer to the "positive" distribution ... - Using PU-learning as a motivation seems a bit weak considering the (sophisticated) optimal transport methods developed in the article. PU-learning is already fairly well understood by using more simple ideas relying on importance sampling (see e.g. [a] or [b] below). Are there more challenging transfer learning tasks that could be solved using your partial-(G)W approach? --> well answered in the rebuttal: color transfer, point clouds registration, deep learning References: [a] du Plessis et al., "Analysis of learning from positive and unlabeled data" [b] Vogel et al., "Weighted Empirical Risk Minimization: Sample Selection Bias Correction based on Importance Sampling"


Review 3

Summary and Contributions: The paper concerns optimal transport (OT) plans minimizing Wasserstein (W) and Gromov-Wasserstein (GW) distances between the empirical distributions of a pair of samples and their application to Positive-Unlabeled (PU) learning. A transport plan between two samples is represented as a matrix with linear equality constraints. The OT plan is formulated as a solution to an optimization problem over a space of transport matrices. The formulation for the Wasserstein distance is based on a single cost matrix representing the pairwise distance between the points of the two samples, whereas the Gromov-Wasserstein distance is based on two cost matrices representing within sample pairwise distances. The paper’s main contribution concerns partial optimal transport plans (for both W and GW distances) that constraint the mass transported between the samples. The formulation relies on the same optimization criteria as for full optimal transport, although with different constraints on the transport matrix. In the case of Wasserstein distance, the formulation is relaxed back to the original constraints by adding two dummy points in the two samples and modifying the cost matrix. In the case of Gromov-Wasserstein (GW) distance, a Frank-Wolfe based algorithm is derived to solve the optimization problem. PU learning is set as a partial OT problem with an additional regularization term. The approach is tested empirically with comparisons to existing PU learning methods.

Strengths: The paper contains novel algorithms on learning partial optimal transport and technically sound theory to support the algorithms. The application to PU learning is intuitive and it is first approach to addresses domain adaptation with different features in the PU context.

Weaknesses: Overall: When applied to PU learning, the theoretical implications are not well understood and an important state of the art method is not compared with. Assuming the class prior to be known is a significant limitation. Furthermore, the time complexity of the algorithm might make it impractical. The paper is difficult to read and significant effort should be spent to simplify the notation, make the paper self contained and wherever possible state the meaning and implication of formulas and constraints. Details: 1) The partial OT formulation enforces that the mass transported back and forth between the two samples is equal. The way PU learning problem is set up, with p_i = 1/n and q_i = 1/m as Partial OT seems to be suboptimal. The mass \pi (positives) from the unlabeled sample gets transported to only mass \pi of the labeled sample. However, ideally it should get transported to the entirety of the labeled sample. If p_i is defined as 1/(n\pi), then the positives in the unlabeled would correspond to the mass of 1 and it can be transported to the entirety of the labeled sample. 2) An important PU method is not included in the comparisons [3]. 3) The algorithms for estimation of the class prior are only suitable for the unbiased PU setting. Assuming that the class prior can be reliably estimated in the biased case is not realistic. Some experiments on the sensitivity to class prior should be conducted. 4) Solving PU learning with bias is an intractable problem in general. That’s why most algorithms for biased PU learning make an assumption for bias first and then derive a method specific to handle that bias. What kind of bias does the proposed method handle? What prevents the method from learning the biased positives from the unlabeled data? It the colored MNIST experiment was designed with the unlabeled data containing both red and green colored digits and the prior was misspecified or set closer to the proportion of the green positives in the unlabeled data, would the partial-GW method assign a higher score to the red positives compared to negatives? Maybe simulating a one dimensional biased PU dataset can be used to illustrate the debiasing effects of the proposed method. 5) The text following equation 6, including the introduction of \alpha, regularization, is quite challenging to follow. Please attempt to give more intuition and make the paper self contained. g and the notation T(i, I_g) is not defined. 6) Would the partial-GW method still work if the negatives were obtained by adding a constant to the positives (think one dimensional data)? The pairwise distances between the negatives would be similar to the pairwise distances between the positives. Is it conceivable that the labeled positives are equally likely to be transported to the negatives or the positives in the unlabeled data? [3] Kiryo R, Niu G, Du Plessis MC, Sugiyama M. Positive-unlabeled learning with non-negative risk estimator. InAdvances in neural information processing systems 2017 (pp. 1675-1685). ========= After Rebuttal========= I still find the section on the regularization unclear. The clarification provided by the author is related to a typo, but not about regularization. If accepted the authors must spend significant effort in making this clear.

Correctness: Yes

Clarity: 1) The equation after line 76 suggests that the number of points and the number of bins are the same. This is counter intuitive since usually a histogram has significantly fewer bins than the size of the sample. Moreover, Dirac functions are point masses, instead of bins. Furthermore, expressing p and q as a summation is not consistent with them introduced as vectors. 2) The paper will be easier to read if words are used to express the meaning of formulas and the implications of constraints are stated explicitly. For example, in section 2, it would be helpful to say that each row of T sums to p_i and each column sums to q_i. And that the ||p|| = ||q|| is implicitly enforced by the constraints on T, since the sum of p_i’s and sum of q_i’s are both enforced to be equal to the sum of all entries of T. 3) Authors should spend effort in improving notations. For example, p_i and q_i can be defined in terms of n_U and n_P. There is no need for n and m. p is used for the vector of probability as well as exponent of the distance. This can be confusing. 5) After stating their choice for p_i and q_i in PU learning, it would help if they provided the rationale behind their choice. 6) The authors claim that the partial-W is relaxed to the full OT problem which is unconstrained. However, the full OT problem has its own set of linear equality.

Relation to Prior Work: yes

Reproducibility: Yes

Additional Feedback: 1) Important references on class proportion estimation are missing [1,2]. 2) How is the PU classifier obtained from the optimal transport? 3) Please comment on the time complexity of the algorithms. [1] Jain S, White M, Trosset MW, Radivojac P. Nonparametric semi-supervised learning of class proportions. arXiv preprint arXiv:1601.01944. 2016 Jan 8 [2] Zeiberg D, Jain S, Radivojac P. Fast Nonparametric Estimation of Class Proportions in the Positive-Unlabeled Classification Setting. InAAAI 2020 (pp. 6729-6736).


Review 4

Summary and Contributions: 1. The paper proposes an algorithm to solve Wasserstein and Gromov-Wasserstein problems. 2. The proposed method can be applied to solve positive-unlabeled learning problem, and the experimental results demonstrate the effectiveness of the proposed method.

Strengths: 1. The paper is well-written and easy to follow. 2. The proposed method is novel and can be applied to solve positive-unlabeled learning problem, and the results are OK.

Weaknesses: My main concerns are listed as follows: 1. There are some typos in the manuscript, e.g., in Abstract, "betwenn". 2. It is a pity that the authors only perform experiments on positive-unlabeled learning, optimal transport techniques have been used in many applications. More results on other applications such as transfer learning, few-shot learning, or zero-shot learning may be better, with more baseline methods being compared. 3. In recent years, both optimal transport and deep learning are hot research issues. The authors are encourage to explain how to expand the proposed method to integrate with deep learning models. 4. For Figure 1, are the figures generated by real experiments or artificially? If they are artificially generated, can authors conduct some real-world experiments to support the phenomenon occurred in these figures? This would be an important evaluation of the proposed method. 5. When citing literature, the tense of sentences is inconsistent, e.g., "Peyré et al. (2016) proposed" and "Chizat et al. (2018) propose".

Correctness: All the claims, methods and empirical methodology are correct.

Clarity: The paper is well-written.

Relation to Prior Work: Yes. For the computation of partial Wasserstein metrics with exact solutions, previous works have no partial formulation.

Reproducibility: Yes

Additional Feedback: The manuscript addresses the partial Wasserstein and Gromov-Wasserstein problems and proposes exact algorithms to solve them. Also, the authors apply their method to positive-unlabeled learning, the proposed method achieves some experimental results to demonstrate the effectiveness. - pos: 1. The manuscript is well-written and the structure is OK. 2. The proposed method can be applied to positive-unlabeled learning problem, I think may be it is also fit to other applications. - con: 1. There are some typos in the manuscript, e.g., in Abstract, "betwenn". 2. It is a pity that the authors only perform experiments on positive-unlabeled learning, optimal transport techniques have been used in many applications. More results on other applications such as transfer learning, few-shot learning, or zero-shot learning may be better, with more baseline methods being compared. 3. In recent years, both optimal transport and deep learning are hot research issues. The authors are encourage to explain how to expand the proposed method to integrate with deep learning models. 4. For Figure 1, are the figures generated by real experiments or artificially? If they are artificially generated, can authors conduct some real-world experiments to support the phenomenon occurred in these figures? This would be an important evaluation of the proposed method. 5. When citing literature, the tense of sentences is inconsistent, e.g., "Peyré et al. (2016) proposed" and "Chizat et al. (2018) propose".

[Author Response · NeurIPS 2020]

**Author Feedback for paper "Partial Optimal Tranport with applications on Positive-Unlabeled Learning"**

We thank the reviewers for their thoughtful feedbacks. We are encouraged that reviewers found the algorithm to solve the partial-OT problem new (R1, R2, R4) and theoretically sound (R1, R3). Moreover, R3, R4 acknowledge that one contribution of the paper is the use of OT for PU learning problems, with ok performances (R1, R4) and with the first application of domain adaptation for PU (R3). We are glad that R2, R4 found the paper pedagogical, well written and that it clearly explains the contributions (R1). We finally thank the reviewers for their careful reading and we will perform a careful proof-reading to fix all mentioned typos. We now review the different comments raised.

**Other applications of exact partial-OT**

One primary concern of R2 and R4 is the lack of consideration of other applications such as color transfer, point clouds registration or deep learning that have already been tackled in the literature (e.g. `https://hal.archives-ouvertes.fr/hal-02111220`, `https://arxiv.org/abs/1607.05816`). We agree. We were constrained by space and we hence put the emphasis on PU learning, which is a novel application for OT. This also allows us to introduce domain adaptation for PU learning, which is, to our knowledge, a new task. Regarding deep learning models, one could consider using partial-OT to detect out-of-distributions examples such as in `https://arxiv.org/abs/1912.12510` or in open set domain adaptation (`https://arxiv.org/abs/1804.10427`). We propose to clarify this in the paper.

**Details about the running time**

R1 and R3 mention that time complexity is not discussed. It is true and we propose to include the running times in the supplementary and to discuss the complexity of the algorithms (cubic for partial-W, several iterations of partial-W for partial-GW). For information, considering all experiments, the maximum time for running one run of partial-W is less than 1.5s and 5.5s for partial-GW.

**Comments regarding the application of PU learning with partial-OT** We now discuss the comments of R3.

• #1 *The mass $\pi$ (positives) from the unlabeled sample gets transported to only mass $\pi$ of the labeled sample* and #5 *The text following equation 6, including the introduction of $\alpha$, regularization, is quite challenging to follow*. There is indeed a typo in line 175: $q_j = 1/m$ should read $q_j = \pi/m$ (as it is correctly formulated in line 182) – many thanks for spotting it! As such, it allows verifying proposition 2, and when choosing $\alpha = 0$, *all* the labeled samples are transported to a mass $\pi$ of the unlabeled sample. We believe that this misspecification of $q_j$ leads to the difficulty in reading the text and we propose to add in line 179 a discussion about $\alpha = 0$.

• #2 *An important PU method is not included in the comparisons [3]* and #3 *Some experiments on the sensitivity to class prior should be conducted.* Since many of the estimation methods of the prior are biased, we agree that it is important to evaluate the influence of the class prior $\pi$ in a biased setting. Note that, as stated in the conclusion, "we plan to derive an extension of this work to PU learning in which the proportion of positives in the dataset will be estimated in a unified optimal transport formulation" but leave it for future work. We then run some additional experiments by varying class prior as in [3] for the MNIST dataset and propose to add the results in the final version of the paper. Note that the method we propose is *transductive*, hence it leads to reduced performances (less than 1 point for all experiments, using $\pi' = 0.8\pi, 0.9\pi, ..., 1.2\pi$). It is true that we do not compare ourselves to [3]. Instead, we prefer relying on Kato et al. (2019), which is a more recent method and which itself builds upon [3]. As such, we strongly believe that the conclusions relative to Kato et al. will resemble the ones that could get by comparing to [3]. Nevertheless, if requested, the comparison could be added in the final version if the paper is accepted.

• #4 *What kind of bias does the proposed method handle?* and #6 *Would the partial-GW method still work if the negatives were obtained by adding a constant to the positives?* Actually, GW is rotation and translation invariant (or more generally invariant to *isometries*). This is a desirable property, as we would like to match data with similar geometry when working on different or unregistered spaces, but in the particular toy case mentioned in #6, this behavior will indeed not allow the detection of the positives among the unlabeled. In the colored MNIST example, partial-GW identifies the unlabeled positives as they have the same "geometry" than the labeled dataset, even if the proportion of green samples in the unlabeled is the same than the prior of the positives (one could expect the method to wrongly label the green ones inside **Unl** as positives). We propose to clarify the invariances of GW in the final version of the paper, then the type of bias that can be handled, together with a deeper discussion of GW as suggested by R2.

**On the notations and clarity**

R3 states that *Authors should spend effort in improving notations* and *the paper will be easier to read if words are used to express the meaning of formulas*. We faced two difficulties: i) space limitation, ii) applying OT for PU learning, two communities which have different notations. As such, we choose to stick to the usual notations of OT (in which $\boldsymbol{p}$ and $\boldsymbol{q}$ usually denote the distributions of $n$ and $m$ bins with the same number of bins as points etc.) and we adjust the ones related to the PU learning, leading to potentially disturbing notations (see comment of R2 about $p$). We make our best to keep the notations consistent, and we try to avoid misunderstandings as much as possible (e.g. by stating **Pos** the labeled positive points rather than **P**). We believe this is not an obstacle to the paper comprehension. To ease the reading, we propose to add more details about the most important equations, and emphasize on the text when a careful reading must be done (e.g. when the source data $\boldsymbol{p}$ correspond to the unlabeled set). Comment #1 of R1: line 103 indicates how to set the mass of the dummy point. Regarding the tuning of $\xi$, Proposition 1 illustrates that this parameter does not influence the solution of Partial-W. In practice we set it as zero as stated in Line 202.

[Meta-Review · NeurIPS 2020]

Overall, the reviewers were quite satisfied with the paper, which provides a conditional gradient algorithm to solve a Gromov-partial Wasserstein problem and theory and experiments to support the algorithms. However I, the area chair, and the senior area chair have discussed this paper in detail and we have a more mitigated opinion. I particular we found several claims to be misleading and they should be changed for the final version to be accepted : - "but when it comes with exact solutions, almost no partial formulation of neither Wasserstein nor Gromov-Wasserstein are available yet." This is not true (partial OT is a standard linear program) so this claim must be removed; - The authors should not claim (such as in the current conclusion) to have introduced the "dummy" point technique for partial optimal transport, which is classical (R#1 gave an example of reference but this is common practice, see e.g. this other reference Pele, Werman, ICCV 2009 "Fast and Robust Earth Mover’s Distances"). Overall, the contributions from the algorithmic optimal transport point of view are rather minor, and we advise the authors to emphasize more on their application to PU learning instead. Other remarks: - please check the grammar (in particular the abstract has several mistakes) - the paper by Solomon et al 2016 "Entropic Metric Alignment for Correspondence Problems" is very relevant in paragraph 2 of the introduction; - what do you mean by "xi" is bounded? (a fixed scalar is always bounded); - the broader impact section is not meant to discuss technical facts, but rather should be about the potential societal impact of the work, if any (see author's guidelines).